# Provably Safe Reinforcement Learning with Step-wise Violation Constraints

**Nuoya Xiong**[1,*], **Yihan Du**[2,†], **Longbo Huang**[1,‡,*]

[1]Institute for Interdisciplinary Information Sciences, Tsinghua University
[2]University of Illinois at Urbana-Champaign
*nuoyaxiong@gmail.com
†yihandu@illinois.edu
‡longbohuang@tsinghua.edu.cn

## Abstract

We investigate a novel safe reinforcement learning problem with step-wise violation constraints. Our problem differs from existing works in that we focus on stricter step-wise violation constraints and do not assume the existence of safe actions, making our formulation more suitable for safety-critical applications that need to ensure safety in all decision steps but may not always possess safe actions, e.g., robot control and autonomous driving. We propose an efficient algorithm SUCBVI, which guarantees an $\widetilde{\mathcal{O}}(\sqrt{ST})$ or gap-dependent $\widetilde{\mathcal{O}}(S/\mathcal{C}_{\text{gap}} + S^2AH^2)$ step-wise violation and an $\widetilde{\mathcal{O}}(\sqrt{H^3SAT})$ regret. Lower bounds are provided to validate the optimality in both violation and regret performance with respect to the number of states $S$ and the total number of steps $T$. Moreover, we further study an innovative safe reward-free exploration problem with step-wise violation constraints. For this problem, we design the algorithm SRF-UCRL to find a near-optimal safe policy, which achieves a nearly state-of-the-art sample complexity $\widetilde{\mathcal{O}}((\frac{S^2AH^2}{\varepsilon} + \frac{H^4SA}{\varepsilon^2})(\log(\frac{1}{\delta}) + S))$, and guarantees an $\widetilde{\mathcal{O}}(\sqrt{ST})$ violation during exploration. Experimental results demonstrate the superiority of our algorithms in safety performance and corroborate our theoretical results.

## 1 Introduction

In recent years, reinforcement learning (RL) (Sutton & Barto, 2018) has become a powerful framework for decision-making and learning in unknown environments. Despite the ground-breaking success of RL in games (Lanctot et al., 2019), recommendation systems (Afsar et al., 2022) and complex tasks in simulation environments (Zhao et al., 2020), most existing RL algorithms focus on optimizing the cumulative reward and do not take into consideration the risk aspect, e.g., the agent runs into catastrophic situations during control. The lack of strong safety guarantees hinders the application of RL to broader safety-critical scenarios such as autonomous driving, robotics and healthcare. For example, for robotic control in complex environments, it is crucial to prevent the robot from getting into dangerous situations, e.g., hitting walls or falling into water pools, at all times.

To handle the safety requirement, a common approach is to formulate safety as a long-term expected violation constraint in each episode. This approach focuses on seeking a policy whose cumulative expected violation in each episode is below a certain threshold. However, for applications where an agent needs to avoid disastrous situations throughout the decision process, e.g., a robot needs to avoid hitting obstacles at each step, merely reducing the long-term expected violation is not sufficient to guarantee safety.

---

*Corresponding author.

37th Conference on Neural Information Processing Systems (NeurIPS 2023).

Motivated by this fact, we investigate safe reinforcement learning with a more fine-grained constraint, called *step-wise* violation constraint, which aggregates all nonnegative violations at each step (no offset between positive and negative violations permitted). We name this problem Safe-RL-SW. Our step-wise violation constraint differs from prior expected violation constraint (Wachi & Sui, 2020; Efroni et al., 2020b; Kalagarla et al., 2021) in two aspects: (i) Minimizing the step-wise violation enables the agent to learn an optimal policy that avoids unsafe regions deterministically, while reducing the expected violation only guarantees to find a policy with low expected violation, instead of a per-step zero-violation policy. (ii) Reducing the aggregated nonnegative violation allows us to have a risk control for each step, while a small cumulative expected violation can still result in a large cost at some individual step and cause danger, if other steps with smaller costs offset the huge cost.

Our problem faces two unique challenges. First, the step-wise violation requires us to guarantee a small violation at each step, which demands very different algorithm design and analysis from that for the expected violation (Wachi & Sui, 2020; Efroni et al., 2020b; Kalagarla et al., 2021). Second, in safety-critical scenarios, the agent needs to identify not only unsafe states but also potentially unsafe states, which are states that may appear to be safe but will ultimately lead to unsafe regions with a non-zero probability. For example, a self-driving car needs to learn to slow down or change directions early, foreseeing the potential danger in advance in order to ensure safe driving (Thomas et al., 2021). Existing safe RL works focus mainly on the expected violation (Wachi & Sui, 2020; Liu et al., 2021b; Wei et al., 2022), or requiring some other assumptions such as imposing the prior knowledge of a safe action for each state (Amani et al., 2021). Moreover, many previous works also require strong assumptions that exclude discrete tabular MDP (Amani et al., 2021; Wachi et al., 2021; Wang et al., 2023) which is considered in our paper. Hence, techniques in previous works cannot be applied to handle step-wise violations. More detailed comparisons are provided in Section 2.

To systematically handle these two challenges, we formulate safety as an unknown cost function for each state without assuming safe actions, and consider minimizing the step-wise violation instead of the expected violation. We propose a general algorithmic framework called **S**afe **UCBVI** (SUCBVI). Specifically, in each episode, we first estimate the transition kernel and cost function in an optimistic manner, tending to regard a state as safe at the beginning of learning. After that, we introduce novel dynamic programming to identify potentially unsafe states and determine safe actions, based on our estimated transition and costs. Finally, we employ the identified safe actions to conduct value iteration. This mechanism can adaptively update dangerous regions, and help the agent plan for the future, which keeps her away from all states that may lead to unsafe states. As our estimation becomes more accurate over time, the safety violation becomes smaller and eventually converges to zero. Note that without the assumption of safe actions, the agent knows nothing about the environment at the beginning. Thus, it is impossible for her to achieve an absolute zero violation. Nevertheless, we show that SUCBVI can achieve a sub-linear $\widetilde{\mathcal{O}}(\sqrt{ST})$ cumulative violation or an $\widetilde{\mathcal{O}}(S/\mathcal{C}_{\text{gap}} + S^2AH^2)$ gap-dependent violation that is independent of $T$. This violation implies that as the RL game proceeds, the agent eventually learns how to avoid unsafe states. We also provide a matching lower bound to demonstrate the optimality of SUCBVI in both violation and regret.

Furthermore, we apply our step-wise safe RL framework to the reward-free exploration (Jin et al., 2020) setting. In this novel safe reward-free exploration, the agent needs to guarantee small step-wise violations during exploration, and also output a near-optimal safe policy. Our algorithm achieves $\varepsilon$ cumulative step-wise violation during exploration, and also identifies a $\varepsilon$-optimal and safe policy. See the definition of $\varepsilon$-optimal and $\varepsilon$-safe policy in Section 6.1. Another interesting application of our framework is safe zero-sum Markov games, which we discuss in Appendix B.

The main contributions of our paper are as follows.

- We formulate the safe RL with step-wise violation constraint problem (Safe-RL-SW), which models safety as a cost function for states and aims to minimize the cumulative step-wise violation. Our formulation is particularly useful for safety-critical applications where avoiding disastrous situations at each decision step is desirable, e.g., autonomous driving and robotics.

- We provide a general algorithmic framework SUCBVI, which is equipped with an innovative dynamic programming to identify potentially unsafe states and distinguish safe actions. We establish an $\widetilde{\mathcal{O}}(\sqrt{H^3SAT})$ regret and an $\widetilde{\mathcal{O}}(\sqrt{ST})$ or $\widetilde{\mathcal{O}}(S/\mathcal{C}_{\text{gap}} + S^2AH^2)$ gap-dependent violation guarantees, which exhibits the capability of SUCBVI in attaining high rewards while maintaining small violation.

- We further establish an $\Omega(\sqrt{HST})$ regret and an $\Omega(\sqrt{ST})$ violation lower bounds for Safe-RL-SW. The lower bounds demonstrate the optimality of algorithm SUCBVI in both regret minimization and safety guarantee, with respect to factors $S$ and $T$.

- We consider step-wise safety constraints in the reward-free exploration setting, which is called the Safe-RFE-SW problem. In this setting, we design an efficient algorithm SRF-UCRL, which ensures $\varepsilon$ step-wise violations during exploration and plans a $\varepsilon$-optimal and $\varepsilon$-safe policy for any reward functions with probability at least $1 - \delta$. We obtain an $\widetilde{\mathcal{O}}((\frac{S^2AH^2}{\varepsilon} + \frac{H^4SA}{\varepsilon^2})(\log(\frac{1}{\delta}) + S))$ sample complexity and an $\widetilde{\mathcal{O}}(\sqrt{ST})$ violation guarantee for SRF-UCRL, which shows the efficiency of SRF-UCRL in sampling and danger avoidance even without reward signals. To the best of our knowledge, this work is the first to study the step-wise violation constraint in the RFE setting.

## 2 Related Work

**Safe RL.** Safety is an important topic in RL, which has been extensively studied. The constrained Markov decision process (CMDP)-based approaches handle safety via cost functions, and aim to minimize the expected episode-wise violation, e.g., (Yu et al., 2019; Wachi & Sui, 2020; Qiu et al., 2020; Efroni et al., 2020b; Turchetta et al., 2020; Ding et al., 2020; Singh et al., 2020; Kalagarla et al., 2021; Simão et al., 2021; Ding et al., 2021), or achieve zero episode-wise violation, e.g., (Liu et al., 2021b; Bura et al., 2021; Wei et al., 2022; Sootla et al., 2022). Apart from CMDP-based approaches, there are also other works that tackle safe RL by control-based approaches (Berkenkamp et al., 2017; Chow et al., 2018; Dalal et al., 2018; Wang et al., 2022), policy optimization (Uchibe & Doya, 2007; Achiam et al., 2017; Tessler et al., 2018; Liu et al., 2020; Stooke et al., 2020) and safety shields (Alshiekh et al., 2018).

In recent years, there are also some works studying step-wise violations with additional assumptions. Now we provide detailed comparisons between existing papers with instantaneous constraints. Turchetta et al. (2016); Wachi et al. (2018) propose a GP-based algorithm, which assumes that the transition is deterministic and known, while modeling the reward and cost functions using Gaussian Processes. By using this particular structure, they can infer the safety cost by estimating the parameters. Wachi et al. (2021) assumes the reward and cost functions have a generalized linear structure. Their algorithm explores in a safe space until a time $t^*$ when the agent explores sufficiently. However, the upper bound of the exploring time $t^*$ is not given in their paper. In fact, under the tabular MDP setting $t^*$ can be infinite.

Amani et al. (2021) further considers the reward and cost functions to have a linear structure. It makes two assumptions: (a) There exists a safe action in each state, which prevents the agent from going to a potentially unsafe state. (b) The feature set is a star convex set, which helps them change actions continuously. However, this makes their works infeasible in tabular MDPs: The feature set in tabular MDPs consists of one-hot vectors and is not a star convex set. Hence, the work Amani et al. (2021) cannot solve our problem. Shi et al. (2023) considers safe RL in linear mixture MDPs. Their work also contains assumption (b), making it infeasible in tabular MDPs. Moreover, although they do not have assumption (a), it assumes the transition set $\Delta(s, a)$ is known, which is not needed in our paper. Thus our paper is more challenging since we need to estimate $\Delta(s, a)$ in our algorithm adaptively.

There are some other papers investigating the safety of RL problems. Alshiekh et al. (2018) represents the safe state by the reactive system, and uses shielding to calculate and restrict the agent within a safe trajectory completely. The main difference between their work and our work is that we need to dynamically update the estimated safe state, while they require to know the mechanism and state to calculate the shield. Dalal et al. (2018) considers restricting the safe action by projecting the action into the closest safe actions. They achieve this goal by solving a convex optimization problem on the continuous action set. However, in their paper, they do not consider the situation where a state can have no safe actions. To be more specific, they do not consider the situation when the convex optimization problem has no solutions. Le et al. (2019) considers the decision-making problem with a pre-collected dataset. Turchetta et al. (2020) and Sootla et al. (2022) both consider cumulative cost constraints rather than step-wise constraints. The former uses a teacher for intervention to keep the agent away from the unsafe region, while the latter encourages safe exploration by augmenting a safety state to measure safety during training.

**Reward-free Exploration with Safety.** Motivated by sparse reward signals in realistic applications, reward-free exploration (RFE) has been proposed in Jin et al. (2020), and further developed

in Kaufmann et al. (2021); Ménard et al. (2021). In RFE, the agent explores the environment without reward signals. After enough exploration, the reward function is given, and the agent needs to plan a near-optimal policy based on his knowledge collected in exploration. Safety is also important in RFE: We need to not only guarantee that our outputted policy is safe, but also ensure small violations during exploration.

Recently, Miryoosefi & Jin (2022); Huang et al. (2022) also study RFE with safety constraints. Compared to our paper, Miryoosefi & Jin (2022) only considers the safety constraint after the exploration phase. Huang et al. (2022) differs from our work in the following aspects: (i) They allow different reward functions during exploration and after exploration, and the agent can directly know the true costs during training. In our work, the cost function is the same during and after exploration, but the agent can only observe the noisy costs of a state when she arrives at that state. (ii) They require prior knowledge of safe baseline policies, while we do not need such an assumption. (iii) They consider the zero-expected violation during exploration, while we focus on keeping small step-wise violations.

## 3 The Safe MDP Model

**Episodic MDP.** In this paper, we consider the finite-horizon episodic Markov Decision Process (MDP), represented by a tuple $(\mathcal{S}, \mathcal{A}, H, \mathbb{P}, r)$. Here $\mathcal{S}$ is the state space, $\mathcal{A}$ is the action space, and $H$ is the length of each episode. $\mathbb{P} = \{\mathbb{P}_h : \mathcal{S} \times \mathcal{A} \mapsto \triangle_{\mathcal{S}}\}_{h \in [H]}$ is the transition kernel, and $\mathbb{P}_h(s'|s, a)$ gives the transition probability from $(s, a)$ to $s'$ at step $h$. $r = \{r_h : \mathcal{S} \times \mathcal{A} \mapsto [0, 1]\}_{h \in [H]}$ is the reward function, and $r_h(s, a)$ gives the reward of taking action $a$ in state $s$ at step $h$. A policy $\pi = \{\pi_h : \mathcal{S} \mapsto \mathcal{A}\}_{h \in [H]}$ consists of $H$ mappings from the state space to action space. In each episode $k$, the agent first chooses a policy $\pi^k$. At each step $h \in [H]$, the agent observes a state $s_h^k$, takes an action $a_h^k$, and then goes to a next state $s_{h+1}^k$ with probability $\mathbb{P}_h(s_{h+1}^k \mid s_h^k, a_h^k)$. The algorithm executes $T = HK$ steps. Moreover, the state value function $V_h^\pi(s, a)$ and state-action value function $Q_h^\pi(s, a)$ for a policy $\pi$ can be defined as

$$V_h^\pi(s) := \mathbb{E}_\pi\left[\sum_{h'=h}^H r_{h'}(s_{h'}, \pi_{h'}(s_{h'})) \middle| s_h = s\right],$$

$$Q_h^\pi(s, a) := \mathbb{E}_\pi\left[\sum_{h'=h}^H r_{h'}(s_{h'}, \pi_{h'}(s_{h'})) \middle| s_h = s, a_h = a\right].$$

**Safety Constraint.** To model unsafe regions in the environment, similar to Wachi & Sui (2020); Yu et al. (2022), we define a safety cost function $c : \mathcal{S} \mapsto [0, 1]$. Let $\tau \in [0, 1]$ denote the safety threshold. A state is called *safe* if $c(s) \leq \tau$, and called *unsafe* if $c(s) > \tau$. Similar to Efroni et al. (2020b); Amani et al. (2021), when the agent arrives in a state $s$, she will receive a cost signal $z(s) = c(s) + \zeta$, where $\zeta$ is an independent, zero-mean and 1-sub-Gaussian noise. Denote $(x)_+ = \max\{x, 0\}$. The violation in state $s$ is defined as $(c(s) - \tau)_+$, and the cumulative step-wise violation till episode $K$ is

$$C(K) = \sum_{k=1}^K \sum_{h=1}^H (c(s_h^k) - \tau)_+. \tag{1}$$

Eq. (1) represents the accumulated step-wise violation during training. When the agent arrives in state $s_h^k$ at step $h$ in episode $k$, she will suffer violation $(c(s_h^k) - \tau)_+$. This violation setting is significantly different from the previous CMDP setting (Qiu et al., 2020; Ding et al., 2020; Efroni et al., 2020b; Ding et al., 2021; Wachi & Sui, 2020; Liu et al., 2021a; Kalagarla et al., 2021). They study the *episode-wise expected violation* $C'(K) = \sum_{k=1}^K \left(\mathbb{E}\left[\sum_{h=1}^H c(s_h^k, a_h^k)\right] - \mu\right)$. There are also some papers (Efroni et al., 2020a; Simão et al., 2021) considering a stricter constraint named *episode-wise clipped expected violation*: $C''(K) = \sum_{k=1}^K \left(\mathbb{E}\left[\sum_{h=1}^H c(s_h^k, a_h^k)\right] - \mu\right)_+$. Compared to the episode-wise violation (including expected violation and clipped expected violation), our step-wise violation has two main differences: (i) First, the episode-wise violation constraints allow the agent to get into unsafe states occasionally. Instead, the step-wise violation constraint forces the agent to stay in safe regions at all times. (ii) Second, in the episode-wise constraints, the average violation

$\mathbb{E}[c(s_h, a_h)] - \mu/H$ at step $h$ is allowed to be positive or negative, and they can cancel out in one episode to achieve $\mathbb{E}[\sum_{h=1}^{H} c(s_h^k, a_h^k)] \leq \mu$. Instead, we consider a *nonnegative* function $(c(s) - \tau)_+$ at each step in our step-wise violation, which imposes a stricter constraint.

Define $\mathcal{U} := \{s \in \mathcal{S} \mid c(s) > \tau\}$ as the set of all *unsafe states*. Let $\iota = \{s_1, a_1, \cdots, s_H, a_H\}$ denote a trajectory. Since a feasible policy needs to satisfy the constraint at each step, we define the set of feasible policies as $\Pi = \{\pi \mid \Pr\{\exists h \in [H], s_h \in \mathcal{U} \mid \iota \sim \pi\} = 0\}$. The feasible policy set $\Pi$ consists of all policies under which one never reaches any unsafe state in an episode.

**Learning Objective.** In this paper, we consider the regret minimization objective. Specifically, define $\pi^* = \operatorname{argmax}_{\pi \in \Pi} V_1^\pi$, $V^* = V^{\pi^*}$ and $Q^* = Q^{\pi^*}$. The regret till $K$ episodes is then defined as

$$R(K) = \sum_{t=1}^{K} (V_1^*(s_1) - V_1^{\pi^k}(s_1)),$$

where $\pi^k$ is the policy taken in episode $k$. Our objective is to minimize $R(K)$ to achieve a good performance, and minimize the violation $C(K)$ to guarantee the safety at the same time.

# 4 Safe RL with Step-wise Violation Constraints

## 4.1 Assumptions and Problem Features

Before introducing our algorithms, we first state the important assumptions and problem features for Safe-RL-SW.

Suppose $\pi$ is a feasible policy. Then, if we arrive at $s_{H-1}$ at step $H - 1$, $\pi$ needs to select an action that guarantees $s_H \notin \mathcal{U}$. Define the *transition set* $\Delta_h(s, a) = \{s' \mid \mathbb{P}_h(s' \mid s, a) > 0\}$ for any $\forall (s, a, h) \in \mathcal{S} \times \mathcal{A} \times [H]$, which represents the set of possible next states after taking action $a$ in state $s$ at step $h$. Then, at the former step $H - 1$, the state $s$ is potentially unsafe if it satisfies that $\Delta_{H-1}(s, a) \cap \mathcal{U} \neq \emptyset$ for all $a \in \mathcal{A}$. (i.e., no matter taking what action, there is a positive probability of transitioning to an unsafe next state). Therefore, we can recursively define the set of *potentially unsafe states at step $h$* as

$$\mathcal{U}_h = \mathcal{U}_{h+1} \cup \{s \mid \forall a \in \mathcal{A}, \Delta_h(s, a) \cap \mathcal{U}_{h+1} \neq \emptyset\}, \tag{2}$$

where $\mathcal{U}_H = \mathcal{U}$. Intuitively, if we are in a state $s_h \in \mathcal{U}_h$ at step $h$, no action can be taken to completely avoid reaching potentially unsafe states $s_{h+1} \in \mathcal{U}_{h+1}$ at step $h + 1$. Thus, in order to completely prevent from getting into unsafe states $\mathcal{U}$ throughout all steps, one needs to avoid potentially unsafe states in $\mathcal{U}_h$ at step $h$. From the above argument, we have that $s_1 \notin \mathcal{U}_1$ is equivalent to the existence of feasible policies. The detailed proof is provided in Appendix D. Thus, we make the following necessary assumption.

**Assumption 4.1** (Existence of feasible policies). The initial state $s_1$ satisfies $s_1 \notin \mathcal{U}_1$.

For any $s \in \mathcal{S}$ and $h \in [H - 1]$, we define the set of safe actions for state $s$ at step $h$ as

$$A_h^{safe}(s) = \{a \in \mathcal{A} \mid \Delta_h(s, a) \cap \mathcal{U}_{h+1} = \emptyset\}, \tag{3}$$

and let $A_H^{safe}(s) = \mathcal{A}$. $A_h^{safe}(s)$ stands for the set of all actions at step $h$ which will not lead to potentially unsafe states in $\mathcal{U}_{h+1}$. Here, $\{\mathcal{U}_h\}_{h \in [H]}$ and $\{A_h^{safe}(s)\}_{h \in [H]}$ are defined by dynamic programming: If we know sets of all possible next state $\{\Delta_h(s, a)\}_{h \in [H]}$ and unsafe state set $\mathcal{U} = \mathcal{U}_H$, we can calculate all potentially unsafe state sets $\{\mathcal{U}_h\}_{h \in [H]}$ and safe action sets $\{A_h^{safe}(s)\}_{h \in [H]}$, and choose feasible policies to completely avoid unsafe states.

## 4.2 Algorithm SUCBVI

Now we present our main algorithm **S**afe **UCBVI** (SUCBVI), which is based on previous classic RL algorithm UCBVI (Azar et al., 2017), and equipped with a novel dynamic programming to identify potentially unsafe states and safe actions. The pseudo-code is shown in Algorithm 1. First, we provide some intuitions about how SUCBVI works. At each episode, we first estimate all the unsafe states based on the historical data. Then, we perform a dynamic programming procedure introduced

---

**Algorithm 1** SUCBVI

---
1: Initialize: $\Delta_h^1(s,a) = \emptyset$, $N_h^1(s,a) = N_h^1(s,a,s') = 0$ for all $s \in \mathcal{S}$, $a \in \mathcal{A}$, $s' \in \mathcal{S}$ and $h \in [H]$.
2: **for** $k = 1, 2, \cdots, K$ **do**
3:     Update the optimistic estimates of cost.
4:     Update the empirical cost $\hat{c}(s)$ and calculate $\bar{c}(s) = \hat{c}(s) - \beta(N^k(s), \delta)$ for all $s \in \mathcal{S}$. ▷
5:     Define $\mathcal{U}_H^k = \{s \mid \bar{c}(s) > \tau\}$ and $\mathcal{U}_h^k = \mathcal{U}_{h+1}^k \cup \{s \mid \forall a \in A, \Delta_h^k(s,a) \cap \mathcal{U}_{h+1}^k \neq \emptyset\}$ for all
    $h \in [H]$ recursively. Calculate $\{A_h^{k,safe}(s)\}_{h \in [H]}$ by Eq. (3) with $\{\mathcal{U}_h^k\}_{h \in [H]}$.
    ▷ Perform value iteration with previous estimates.
6:     **for** $h = H, H-1, \cdots, 1$ **do**
7:       **for** $s \in \mathcal{S}$ **do**
8:         **for** $a \in \mathcal{A}$ **do**
9:           Compute $\hat{\mathbb{P}}_h^k(s' \mid s,a) = \frac{N_h^k(s,a,s')}{N_h^k(s,a)}$.
10:           $Q_h^k(s,a) = \min\{H, r_h(s,a) + \sum_{s'} \hat{\mathbb{P}}_h^k(s' \mid s,a)V_{h+1}^k(s') + \alpha(N_h^k(s,a))\}$.
11:         **end for**
12:         **if** $s \notin \mathcal{U}_h^k$ **then**
13:           $V_h^k(s) = \max_{a \in A_h^{k,safe}(s)} Q_h^k(s,a), \pi_h^k(s) = \arg\max_{a \in A_h^{k,safe}(s)} Q_h^k(s,a)$ .
14:         **else**
15:           $V_h^k(s) = \max_{a \in \mathcal{A}} Q_h^k(s,a), \pi_h^k(s) = \arg\max_{a \in \mathcal{A}} Q_h^k(s,a)$.
16:         **end if**
17:       **end for**
18:     **end for**
19:     **for** $h = 1, 2, \cdots, H$ **do**
20:       Take action $a_h^k = \pi_h^k(s_h^k)$ and observe state $s_{h+1}^k$.
      ▷ Update the estimates of $\Delta(s,a)$.
21:       $\Delta_h^{k+1}(s_h^k, a_h^k) = \Delta_h^k(s_h^k, a_h^k) \cup \{s_{h+1}^k\}$. Increase $N_h^{k+1}(s_h^k, a_h^k), N_h^{k+1}(s_h^k, a_h^k, s_h^k)$ by 1.
22:     **end for**
23: **end for**

---

in Section 4.1 and calculate the safe action set $A_h^{safe}(s)$ for each state $s$. Then, we perform value iteration in the estimated safe action set. As the estimation becomes more accurate, SUCBVI will eventually avoid potentially unsafe states and achieve both sublinear regrets and violations.

Now we begin to introduce our algorithm. In the beginning, we initialize $\Delta_h(s,a) = \emptyset$ for all $(h,s,a) \in [H] \times \mathcal{S} \times \mathcal{A}$. It implies that the agent considers all actions to be safe at first, because no action will lead to unsafe states from the agent's perspective. In each episode, we first estimate the empirical cost $\hat{c}(s)$ based on historical cost feedback $z(s)$, and regard state $s$ as safe if $\bar{c}(s) = \hat{c}(s) - \beta > \tau$ for some bonus term $\beta$, which aims to guarantee $\bar{c}(s) \leq c(s)$ (Line 4). Then, we calculate the estimated unsafe state set $\mathcal{U}_H^k$, which is a subset of the true unsafe state set $\mathcal{U} = \mathcal{U}_H$ with a high probability by optimism. With $\mathcal{U}_H^k$ and $\Delta_h^k(s,a)$, we can estimate potentially unsafe state sets $\mathcal{U}_h^k$ for all $h \in [H]$ by Eq. (2) recursively.

Then, we perform value iteration to compute the optimistically estimated optimal policy. Specifically, for any hypothesized safe state $s \notin \mathcal{U}_h^k$, we update the greedy policy on the estimated safe actions, i.e., $\pi_h^k(s) = \max_{a \in A_h^{k,safe}(s)} Q_h^k(s,a)$ (Line 13). On the other hand, for any hypothesized unsafe state $s_h \in \mathcal{U}_h$, since there is no action that can completely avoid unsafe states, we ignore safety costs and simply update the policy by $\pi_h^k(s) = \max_{a \in \mathcal{A}} Q(s,a)$ (Line 15). After that, we calculate the estimated optimal policy $\pi^k$ for episode $k$, and the agent follows $\pi^k$ and collects a trajectory. Then, we update $\Delta_h^{k+1}(s,a)$ by incorporating the observed state $s_{h+1}^k$ into the set $\Delta_h^k(s_h^k, a_h^k)$. Under this updating rule, it holds that $\Delta_h^k(s,a) \subseteq \Delta_h(s,a)$ for all $(s,a) \in \mathcal{S} \times \mathcal{A}$.

The performance of Algorithm 1 is summarized below in Theorem 4.2.

**Theorem 4.2.** *Let* $\alpha(n,\delta) = 7H\sqrt{\frac{\ln(5SAHK/\delta)}{n}}$ *and* $\beta(n,\delta) = \sqrt{\frac{2}{n}\log(SK/\delta)}$. *With probability at least* $1 - \delta$, *the regret and step-wise violation of Algorithm 1 are bounded by*

$$R(K) = \widetilde{\mathcal{O}}(\sqrt{H^2 SAT}), \qquad C(K) = \widetilde{\mathcal{O}}(\sqrt{ST} + S^2 AH^2).$$

*Moreover, if $\mathcal{C}_{\text{gap}} \triangleq \min_{s \in \mathcal{U}}(c(s) - \tau)_+ > 0$, we have $C(K) = \widetilde{\mathcal{O}}(S/\mathcal{C}_{\text{gap}} + S^2 AH^2)$.*

Theorem 4.2 shows that SUCBVI achieves both sublinear regret and violation. Moreover, when all the unsafe states have a large cost compared to the safety threshold $\tau$, i.e., $\mathcal{C}_{\text{gap}} = \min_{s \in \mathcal{U}}(c(s) - \tau)_+ > 0$ is a constant, we can distinguish the unsafe states easily and get a constant violation. In particular, when $\tau = 1$, Safe-RL-SW degenerates to the unconstrained MDP and Theorem 4.2 maintains the same regret $\widetilde{\mathcal{O}}(\sqrt{H^2 SAT})$ as UCBVI-CH (Azar et al., 2017), while CMDP algorithms (Efroni et al., 2020b; Liu et al., 2021a) suffer a larger $\widetilde{\mathcal{O}}(\sqrt{H^3 S^3 AT})$ regret.

We provide the analysis idea of Theorem 4.2 here. First, by the updating rule of $\Delta_h^k(s, a)$, we show that $\mathcal{U}_h^k$ and $A_h^{k,safe}(s)$ have the following crucial properties: $\mathcal{U}_h^k \subseteq \mathcal{U}_h$, $A_h^{safe}(s) \subseteq A_h^{k,safe}(s)$. Based on this property, we can prove that $Q_h^*(s, a) \leq Q_h^k(s, a)$ and $V_h^*(s) \leq V_h^k(s)$ for all $(s, a)$, and then apply the regret decomposition techniques to derive the regret bound.

Recall that $\pi^k$ is a feasible policy with respect to the estimated unsafe state set $\mathcal{U}_H^k$ and transition set $\{\Delta_h^k(s, a)\}_{h \in [H]}$. If the agent takes policy $\pi^k$ and the transition follows $\Delta_h^k(s, a)$, i.e., $s_{h+1}^k \in \Delta_h^k(s, a)$, the agent never arrive any estimated unsafe state $s \in \mathcal{U}_H^k$ in this episode. Hence the agent will suffer at most an $\widetilde{\mathcal{O}}(\sqrt{ST})$ step-wise violation or an $\widetilde{\mathcal{O}}(S/\mathcal{C}_{\text{gap}} + S^2 AH^2)$ gap-dependent bounded violation. Yet, the situation $s_{h+1} \in \Delta_h^k(s, a)$ does not always hold for all $h \in [H]$. In the case when $s_{h+1}^k \notin \Delta_h^k(s, a)$, we add the newly observed state $s_{h+1}^k$ into $\Delta_h^{k+1}(s_h^k, a_h^k)$ (Line 21). We can show that this case appears at most $S^2 AH$ times, and thus incurs $O(S^2 AH^2)$ additional violation. Combining the above two cases, the total violation can be upper bounded.

# 5   Lower Bounds for Safe-RL-SW

In this section, we provide a matching lower bound for Safe-RL-SW in Section 4. The lower bound shows that if an algorithm always achieves a sublinear regret in Safe-RL-SW, it must incur an $\Omega(\sqrt{ST})$ violation. This result matches our upper bound in Theorem 4.2, showing that SUCBVI achieves the optimal violation performance.

**Theorem 5.1.** *If an algorithm has an expected regret $\mathbb{E}_\pi[R(K)] \leq \frac{HK}{24}$ for all MDP instances, there exists an MDP instance in which the algorithm suffers expected violation $\mathbb{E}_\pi[C(K)] = \Omega(\sqrt{ST})$.*

Now we validate the optimality in terms of regret. Note that if we do not consider safety constraints, the lower bound for classic RL (Osband & Van Roy, 2016) can be applied to our setting. Thus, we also have an $\Omega(\sqrt{T})$ regret lower bound. To understand the essential hardness brought by safety constraints, we further investigate whether safety constraints will lead to an $\Omega(\sqrt{T})$ regret, given that we can achieve an $o(\sqrt{T})$ regret on some good instances without the safety constraints.

**Theorem 5.2.** *For any $\alpha \in (0, 1)$, there exists a parameter $n$ and $n$ MDPs $M_1, \ldots, M_n$ satisfying that:*

*1. If we do not consider any constraint, there is an algorithm that achieves an $\widetilde{\mathcal{O}}(T^{(1-\alpha)/2})$ regret compared to the unconstrained optimal policy on all MDPs.*

*2. If we consider the safety constraint, any algorithm with a $O(T^{1-\alpha})$ expected violation will achieve an $\Omega(\sqrt{HST})$ regret compared to the constrained optimal policy on one of MDPs.*

Intuitively, Theorem 5.2 shows that if one achieves sublinear violation, she must suffer at least an $\Omega(\sqrt{T})$ regret even if she can achieve an $o(\sqrt{T})$ regret without considering constraints. This theorem demonstrates the hardness particularly brought by the step-wise constraint, and corroborates the optimality of our results. Combining with Theorem 5.1, the two lower bounds show an essential trade-off between the violation and performance.

# 6 Safe Reward-Free Exploration with Step-wise Violation Constraints

## 6.1 Formulation of Safe-RFE-SW

In this section, we consider Safe RL in the reward-free exploration (RFE) setting (Jin et al., 2020; Kaufmann et al., 2021; Ménard et al., 2021) called Safe-RFE-SW, to show the generality of our proposed framework. In the RFE setting, the agent does not have access to reward signals and only receives random safety cost feedback $z(s) = c(s) + \zeta$. To impose safety requirements, Safe-RFE-SW requests the agent to keep small safety violations during exploration, and outputs a near-optimal safe policy after receiving the reward function.

**Definition 6.1** (($\varepsilon, \delta$)-optimal safe algorithm for Safe-RFE-SW). An algorithm is ($\varepsilon, \delta$)-optimal safe for Safe-RFE-SW if it outputs the triple $(\hat{\mathbb{P}}, \hat{\Delta}(s, a), \hat{\mathcal{U}}_H)$ such that for any reward function $r$, with probability at least $1 - \delta$,

$$V_1^*(s_1; r) - V_1^{\hat{\pi}^*}(s_1; r) \leq \varepsilon, \qquad \mathbb{E}_\pi \left[ \sum_{h=1}^{H} (c(s_h) - \tau)_+ \right] \leq \varepsilon, \ \forall \pi \in \Pi \qquad (4)$$

where $\hat{\pi}^*$ is the optimal feasible policy with respect to $(\hat{\mathbb{P}}, \hat{\Delta}(s, a), \hat{\mathcal{U}}_H, r)$, $\Pi$ is the set of feasible policies with respect to $(\Delta(s, a), \mathcal{U}_H)$. and $V(s; r)$ is the value function under reward function $r$. We say that a policy is $\varepsilon$-*optimal* if it satisfies the left inequality in Eq. (4) and $\varepsilon$-*safe* if it satisfies the right inequality in Eq. (4). We measure the performance by the number of episodes used before the algorithm terminates, i.e., *sample complexity*. Moreover, the cumulative step-wise violation till episode $K$ is defined as $C(K) = \sum_{k=1}^{K} \sum_{h=1}^{H} (c(s_h^k) - \tau)_+$. In Safe-RFE-SW, our goal is to design an ($\varepsilon, \delta$)-optimal safe algorithm, and minimize both sample complexity and violation.

## 6.2 Algorithm SRF-UCRL

The Safe-RFE-SW problem requires us to consider extra safety constraints for both the exploration phase and final output policy, which needs new algorithm design and techniques compared to previous RFE algorithms. Also, the techniques for Safe-RL-SW in Section 4.2 are not sufficient for guaranteeing the safety of output policy, because SUCBVI only guarantees a step-wise violation during exploration.

We design an efficient algorithm **S**afe **RF-UCRL** (SRF-UCRL), which builds upon previous RFE algorithm RF-UCRL (Kaufmann et al., 2021). SRF-UCRL distinguishes potentially unsafe states and safe actions by backward iteration, and establishes a new uncertainty function to guarantee the safety of output policy. Algorithm 2 illustrates the procedure of SRF-UCRL. Specifically, in each episode $k$, we first execute a policy $\pi_h^k$ computed from previous episodes, and then update the estimated next state set $\{\Delta_h^k(s, a)\}_{h \in [H]}$ and unsafe state set $\mathcal{U}_H^k$ by optimistic estimation. Then, we use Eq. (2) to calculate the unsafe state set $\mathcal{U}_h$ for all steps $h \in [H]$. After that, we update the uncertainty function $\overline{W}^k$ defined in Eq. (5) below and compute the policy $\pi^{k+1}$ that maximizes the uncertainty to encourage more exploration in the next episode.

Now we provide the definition of the *uncertainty function*, which measures the estimation error between the empirical MDP and true MDP. For any safe state-action pair $s \notin \mathcal{U}_h, a \in A_h^{k,safe}(s)$, we define

$$\overline{W}_h^k(s, a) = \min \left\{ H, M(N_h^k(s, a), \delta) + \sum_{s'} \hat{\mathbb{P}}_h^k(s' \mid s, a) \max_{b \in A_{h+1}^{k,safe}(s')} \overline{W}_{h+1}^k(s', b) \right\}. \qquad (5)$$

where $M(N_h^k(s, a), \delta) = 2H \sqrt{\frac{2\gamma(N_h^k(s,a), \delta)}{N_h^k(s,a)}} + \frac{SH\gamma(N_h^k(s,a), \delta)}{N_h^k(s,a)}$, and $\gamma(n, \delta)$ is a logarithmic term that is formally defined in Theorem 6.2. For other state-action pairs $(s, a)$, we relax the restriction of $b \in \mathcal{A}_{h+1}^{k,safe}(s')$ to $b \in \mathcal{A}$ in the max function. Our algorithm stops when $\overline{W}_1^K(s_1, \pi^K(s_1))$ shrinks to within $\varepsilon/2$. Compared to previous RFE works (Kaufmann et al. (2021); Ménard et al. (2021)), our uncertainty function has two distinctions. First, for any safe state-action pair $(s, a) \in (\mathcal{S} \setminus \mathcal{U}_h, A_h^{k,safe}(s))$, Eq. (5) considers only safe actions $A_h^{k,safe}(s)$, which guarantees that the agent focuses on safe policies. Second, Eq. (5) incorporates another term $(SH\gamma(N_h^k(s, a), \delta)/N_h^k(s, a))$ to control the expected violation for feasible policies. Now we present our result for Safe-RFE-SW.

---

**Algorithm 2** SRF-UCRL

---

1: Initialize: $k = 1$, $\overline{W}_h^0(s, a) = H$, $\pi_h^1(s) = a_1$ for all $s \in \mathcal{S}$, $a \in \mathcal{A}$ and $h \in [H]$.
2: **while** $\overline{W}_h^{k-1}(s_1, \pi_1^k(s_1)) \leq \varepsilon/2$ **do**
3:     **for** $h = 1, \cdots, H$ **do**
4:         Observe state $s_h^k$. Take action $a_h^k = \pi_h^k(s_h^k)$ and observe $s_{h+1}^k$.
        ▷ Update the optimistic estimate of cost.
5:         Calculate empirical cost $\hat{c}(s)$ and $\bar{c}(s) = \hat{c}(s) - \beta(N^k(s), \delta)$ for all $s \in \mathcal{S}$.
        ▷ Update the estimates of $\Delta_h(s, a)$ and $\mathcal{U}_h$.
6:         Update $\Delta_h^k(s_h^k, a_h^k) = \Delta_h^{k-1}(s_h^k, a_h^k) \cup \{s_{h+1}^k\}$ and $\{\mathcal{U}_h^k\}_{h \in [H]}$ by Eq. (2).
7:         Calculate $A_h^{k,safe}(s)$ for all $s \in \mathcal{S}$ by Eq. (3).
8:     **end for**
9:     Update $N_h^k(s, a, s')$, $N_h^k(s, a)$ and $\hat{\mathbb{P}}_h^k$ for all $s \in \mathcal{S}$, $a \in \mathcal{A}$, $s' \in \mathcal{S}$ and $h \in [H]$.
10:     Compute $\overline{W}^k$ according to Eq. (5).
    ▷ Calculate the greedy policy.
11:     **for** $h \in [H], s \in \mathcal{S}$ **do**
12:         **if** $A_h^{k,safe}(s) \neq \emptyset$ **then**
13:             $\pi_h^{k+1}(s) = \arg\max_{a \in A_h^{k,safe}(s)} \overline{W}_h^k(s, a)$.
14:         **else**
15:             $\pi_h^{k+1}(s) = \arg\max_{a \in A} \overline{W}_h^k(s, a)$.
16:         **end if**
17:     **end for**
18:     Set $k = k + 1$.
19: **end while**
20: **return** the tuple $(\hat{\mathbb{P}}_h^{k-1}, \Delta_h^{k-1}(s, a), \mathcal{U}_H^{k-1})$.

---

**Theorem 6.2.** *Let* $\gamma(n, \delta) = 2(\log(2SAH/\delta) + (S - 1)\log(e(1 + n/(S - 1))))$, *Algorithm 2 is a* $(\varepsilon, \delta)$-*PAC algorithm with sample complexity at most*[2]

$$K = \widetilde{\mathcal{O}}\left(\left(\frac{S^2 A H^2}{\varepsilon} + \frac{H^4 SA}{\varepsilon^2}\right)\left(\log\left(\frac{1}{\delta}\right) + S\right)\right),$$

*The step-wise violation of Algorithm 2 during exploration is* $C(K) = \widetilde{\mathcal{O}}(S^2 A H^2 + \sqrt{ST})$.

Compared to previous work (Kaufmann et al., 2021) with an $\widetilde{\mathcal{O}}((H^4 SA/\varepsilon^2)(\log(1/\delta) + S))$ sample complexity, our result has an additional term $\widetilde{\mathcal{O}}((S^2 A H^2/\varepsilon)(\log(1/\delta) + S))$. This term is due to the additional safety requirement for the final output policy, which was not considered in previous RFE algorithms (Kaufmann et al., 2021; Ménard et al., 2021). When $\varepsilon$ and $\delta$ are sufficiently small, the leading term is $\widetilde{\mathcal{O}}((H^4 SA/\varepsilon^2)\log(1/\delta))$, which implies that our algorithm satisfies the safety constraint without suffering additional regret.[3]

### 6.3 Analysis for Algorithm SRF-UCRL

For the analysis of step-wise violation (Eq. (1)), similar to algorithm SUCBVI, algorithm SRF-UCRL estimates the next state set $\Delta_h(s, a)$ and potentially unsafe state set $\mathcal{U}_h$, which guarantees a $\widetilde{\mathcal{O}}(\sqrt{ST})$ step-wise violation. Now we give a proof sketch for the $\varepsilon$-safe property of output policy (Eq. (4)).

First, if $\pi$ is a feasible policy for $(\hat{\mathbb{P}}^k, \Delta^k(s, a), \{\mathcal{U}_h^k\}_{h \in [H]})$ and $s_{h+1} \in \Delta_h^K(s_h, a_h)$ for all $h \in [H]$, the agent who follows policy $\pi$ will only visit the estimated safe states. Since each estimated safe state

---

[2]Here $\widetilde{\mathcal{O}}(\cdot)$ ignores all $\log S, \log A, \log H, \log(1/\varepsilon)$ and $\log(\log(1/\delta))$ terms.
[3]Ménard et al. (2021) improve the result by a factor $H$ and replace $\log(1/\delta) + S$ by $\log(1/\delta)$ via the Bernstein-type inequality. While they do not consider the safety constraints, we believe that similar improvement can also be applied to our framework, without significant changes in our analysis. However, since our paper mainly focuses on tackling the safety constraint for reward-free exploration, we use the Hoeffding-type inequality to keep the succinctness of our statements.

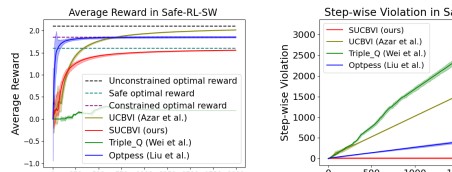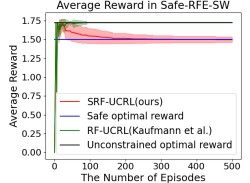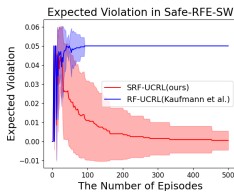

Figure 1: Experimental results for Safe-RL-SW and Safe-RFE-SW. The left two figures show the average rewards and step-wise violations of algorithms SUCBVI, UCBVI (Azar et al., 2017), OptCMDP-bonus (Efroni et al., 2020b), Triple-Q (Wei et al., 2022) and Optpess (Liu et al., 2021a). The right two figures show the reward and expected violation of the policies outputted by algorithms SRF-UCRL and RF-UCRL (Kaufmann et al., 2021).

only suffers a $\mathcal{O}(1/\sqrt{N_t(s_h, \pi_h(s_h))})$ violation, the violation led by this situation is bounded by $\overline{W}_1^k(s, \pi_1(s_1))$. Next, we bound the probability that $s_{h+1} \notin \Delta_h^K(s_h, a_h)$ for some step $h \in [H]$. For any state-action pair $(s, a)$, if there is a probability $\mathbb{P}(s' \mid s, a) \geq \widetilde{\mathcal{O}}(\log(S/\delta)/N_h^K(s, a))$ that the agent transitions to next state $s'$ from $(s, a)$ at step $h$, the state $s'$ is put into $\Delta_h^K(s, a)$ with probability at least $1 - \delta/S$. Then, we can expect that all such states are put into our estimated next state set $\Delta_h^K(s, a)$. Thus, the probability that $s_{h+1} \notin \Delta_h^K(s_h, a_h)$ is no larger than $\widetilde{\mathcal{O}}(S \log(S/\delta)/N_h^K(s, a))$ by a union bound over all possible next states $s'$. Based on this argument, $\overline{W}_h^k(s_h, a_h)$ is an upper bound for the total probability that $s_{h+1} \notin \Delta_h^K(s_h, a_h)$ for some step $h$. This will lead to additional $\overline{W}_1^K(s_1, \pi_1(s_1))$ expected violation. Hence the expected violation of output policy is upper bounded by $2\overline{W}_1^K(s_1, \pi_1^K(s_1)) \leq \varepsilon$. The complete proof is provided in Appendix A.

## 7 Experiments

In this section, we provide experiments for Safe-RL-SW and Safe-RFE-SW to validate our theoretical results. For Safe-RL-SW, we compare our algorithm SUCBVI with a classical RL algorithm UCBVI (Azar et al., 2017) and three state-of-the-art CMDP algorithms OptCMDP-bonus (Efroni et al., 2020b), Optpess (Liu et al., 2021a) and Triple-Q (Wei et al., 2022). For Safe-RFE-SW, we report the average reward in each episode and cumulative step-wise violation. For Safe-RFE-SW, we compare our algorithm SRF-UCRL with a state-of-the-art RFE algorithm RF-UCRL (Kaufmann et al., 2021). We do not plot the regret because unconstrained MDP or CMDP algorithms do not guarantee step-wise violation. Applying them to the step-wise constrained setting can lead to negative or large regret and large violations, making the results meaningless. Detailed experiment setup is in Appendix E.

As shown in Figure 1, the rewards of SUCBVI and SRF-UCRL converge to the optimal rewards under safety constraints (denoted by "safe optimal reward"). In contrast, the rewards of UCBVI and UCRL converge to the optimal rewards without safety constraints (denoted by "unconstrained optimal reward"), and those of CMDP algorithms converge to a policy with low expected violation (denoted by "constrained optimal reward"). For Safe-RFE-SW, the expected violation of output policy of SRF-UCRL converges to zero while that of RF-UCRL does not. This corroborates the ability of SRF-UCRL in finding policies that simultaneously achieve safety and high rewards.

## 8 Conclusion

In this paper, we investigate a novel safe reinforcement learning problem with step-wise safety constraints. We first provide an algorithmic framework SUCBVI to achieve both an $\widetilde{\mathcal{O}}(\sqrt{H^3 SAT})$ regret and an $\widetilde{\mathcal{O}}(\sqrt{ST})$ step-wise or an $\widetilde{\mathcal{O}}(S/\mathcal{C}_{\text{gap}} + S^2 AH^2)$ gap-dependent bounded violation that is independent of $T$. Then, we provide two lower bounds to validate the optimality of SUCBVI in both violation and regret in terms of $S$ and $T$. Further, we extend our framework to the safe RFE with a step-wise violation and provide an algorithm SRF-UCRL that identifies a near-optimal safe policy given any reward function $r$ and guarantees an $\widetilde{\mathcal{O}}(\sqrt{ST})$ violation during exploration.

## Acknowledgements

This work is supported by the Technology and Innovation Major Project of the Ministry of Science and Technology of China under Grant 2020AAA0108400 and 2020AAA0108403 and the Tsinghua Precision Medicine Foundation 10001020109.

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
