# Appendix

## A  Proofs of Main Results

### A.1  Proof of Theorem 4.2

We start the proof with the confidence bound of optimistic estimation:

**Lemma A.1.** *With probability at least $1 - \delta$, the empirical optimistic estimation of cost function $\overline{c}^k(s) = \hat{c}^k(s) - \sqrt{\frac{2}{N^k(s)} \log(SK/\delta)}$ are always less than the true cost $c(s)$ for any episode $k \in [K]$. In other words, let the event be*

$$\mathcal{E}_1 = \left\{ \forall\, k \in [K] \text{ and } s \in \mathcal{S}, \overline{c}^k(s) \leq c(s) \right\}, \tag{6}$$

*then $\Pr(\mathcal{E}_1^c) \leq \delta$.*

*Proof.* Since we observe $c(s) + \varepsilon$ with 1-subgaussian noise $\varepsilon$, by Hoeffding's inequality Hoeffding (1994), for a fixed episode $k$ and state $s$, the statement

$$|\hat{c}^k(s) - c(s)| \leq \sqrt{\frac{2}{N^k(s)} \log\left(\frac{SK}{\delta}\right)}$$

holds with probability at least $1 - \frac{\delta}{SK}$. Taking a union bound on all $s \in \mathcal{S}$ and $k \in [K]$, we have

$$c(s) \geq \hat{c}^k(s) - \sqrt{\frac{2}{N^k(s)} \log\left(\frac{SK}{\delta}\right)} = \overline{c}^k(s),$$

and the Lemma A.1 has been proved. $\qquad\square$

**Lemma A.2.** *Let the event $\mathcal{E}_2$ and $\mathcal{E}_3$ are*

$$\mathcal{E}_2 = \Big\{ \forall\, s, a, s', k, h, |\hat{\mathbb{P}}_h(s' \mid s, a) - \mathbb{P}_h(s' \mid s, a)|$$

$$\leq \sqrt{\frac{2\mathbb{P}_h(s' \mid s, a)}{N_h^k(s, a)} \log\left(\frac{2S^2 AHK}{\delta}\right)} + \frac{2}{3N_h^k(s, a)} \log\left(\frac{2S^2 AHK}{\delta}\right) \Big\}$$

$$\mathcal{E}_3 = \Big\{ \sum_{k=1}^{K}\sum_{h=1}^{H} \Big( P_h(V_{h+1}^k - V_{h+1}^{\pi^k})(s_h^k, a_h^k) - (V_{h+1}^k - V_{h+1}^{\pi^k})(s_{h+1}^k) \Big) \leq \sqrt{\frac{H^3 K}{2} \log(1/\delta)} \Big\}$$

*Then $\Pr(\mathcal{E}_2^c) \leq \delta$, $\Pr(\mathcal{E}_3^c) \leq \delta$.*

*Proof.* This lemma is very standard in analysis of UCBVI algorithm. The $\Pr(\mathcal{E}_2^c)$ can be derived by Bernstein's inequality, and $\Pr(\mathcal{E}_3^c) \leq \delta$ can be derived by Azuma-hoeffding inequality. $\square$

The following lemma shows the optimistic property we discussed in the paper.

**Lemma A.3.** *Under the event $\mathcal{E}_1$, $\mathcal{U}_h^k \subseteq \mathcal{U}_h$ and $A_h^{safe}(s) \subseteq A_h^{k,safe}(s)$ for all $s \in \mathcal{U}_h^k$ and $h \in [H]$.*

*Proof.* Under the event $\mathcal{E}_1$, for all $s \in \mathcal{U}_H^k$, $c(s) \geq \bar{c}(s) > \tau$. So $s \in \mathcal{U}_H^*$. Now from our definition of $\Delta_h^K(s, a)$, we can easily have $\Delta_h^k(s, a) \subseteq \Delta_h(s, a)$ for all step $h \in [H]$ because we add the possible next state one by one into estimated set $\Delta_h^K(s, a)$.

Now we prove the lemma by induction. When $h = H, \mathcal{U}_H^k \subseteq \mathcal{U}_H^*$. Assume that $\mathcal{U}_{h+1}^k \subseteq \mathcal{U}_{h+1}^*$, then

$$A_h^{safe}(s) = \{a \in \mathcal{A} \mid \Delta_h(s, a) \cap \mathcal{U}_{h+1}^* = \emptyset\}$$
$$\subseteq \{a \in \mathcal{A} \mid \Delta_h^K(s, a) \cap \mathcal{U}_{h+1}^* = \emptyset\}$$
$$\subseteq \{a \in \mathcal{A} \mid \Delta_h^K(s, a) \cap \mathcal{U}_{h+1}^k = \emptyset\}$$
$$= A_h^{k,safe}(s),$$

and

$$\mathcal{U}_h^k = \mathcal{U}_{h+1}^k \cup \{s \mid \exists\, a \in \mathcal{A}, \Delta_h^K(s, a) \cap \mathcal{U}_{h+1}^k = \emptyset\}$$
$$\subseteq \mathcal{U}_{h+1}^* \cup \{s \mid \exists\, a \in \mathcal{A}, \Delta_h(s, a) \cap \mathcal{U}_{h+1}^* = \emptyset\}$$
$$\subseteq \mathcal{U}_h.$$

Hence we complete the proof by induction. $\square$

Similar to UCBVI algorithm, we prove that $Q_h^k(s, a)$ and $V_h^k(s)$ are always greater than $Q_h^*(s)$ and $V_h^*(s)$.

**Lemma A.4.** *With probability at least $1 - \delta$, choosing $b(n, \delta) = 7H\sqrt{\frac{\ln(5SAT/\delta)}{n}}$, we will have*

$$Q_h^k(s, a) \geq Q_h^*(s, a), \quad V_h^k(s) \geq V_h^*(s),$$

*for any state $s \in \mathcal{S}$, action $a \in \mathcal{A}$, episode $k$ and the step $h \in [H]$.*

*Proof.* We proof this lemma by induction. First, consider $h = H + 1$, then $Q_{H+1}^*(s, a) = Q_{H+1}^k(s, a) = 0$ and $V_{H+1}^*(s) = V_{H+1}^k(s) = 0$ for all $k \in [K]$. Then our statements hold for $h = H + 1$. Now fixed an episode $k$ and assume this statement holds for $h + 1$. If $Q_h^k(s, a) = H$, our proof is completed. Then assume $Q_h^k(s, a) \leq H$ and

$$Q_h^k(s, a) - Q_h^*(s, a) = (\hat{\mathbb{P}}_h^k)(V_{h+1} - V^*)(s, a) + (\hat{\mathbb{P}}_h^k - \mathbb{P}_h)(V^*)(s, a) + b(N_h^k(s, a))$$
$$\geq (\hat{\mathbb{P}}_h^k - \mathbb{P}_h)(V^*)(s, a) + b(N_h^k(s, a))$$
$$\geq 0$$

where the first inequality is the induction hypothesis, and the second inequality follows naturally by Chernoff-Hoeffding's inequality. Then for $s \in \mathcal{U}_h^k$

$$V_h^k(s) = \max_{a \in \mathcal{A}} Q_h^k(s, a) \geq \max_{a \in \mathcal{A}} Q^*(s, a) \geq V_h^*(s).$$

For $s \notin \mathcal{U}_h^k$, since we know $\mathcal{U}_h \subseteq \mathcal{U}_h^k$, $s \notin \mathcal{U}_h$. Then

$$
\begin{aligned}
V_h^k(s) = \max_{a \in A_h^{k,safe}(s)} Q_h^k(s,a) &\geq \max_{a \in A_h^{k,safe}(s)} Q_h^*(s,a) \\
&\geq \max_{a \in A_h^{safe}(s)} Q_h^*(s,a) \\
&= V_h^*(s).
\end{aligned}
\tag{7}
$$

where Eq. (7) is because $A_h^{safe}(s) \subseteq A_h^{k,safe}(s)$ by our optimistic exploration mechanism. $\qquad \square$

**Regret**  Now we can start to prove our main theorem.

*Proof.* Having the previous analysis, our proof is similar to UCBVI algorithm. Note that the regret can be bounded by UCB:

$$
R(K) = \sum_{k=1}^{K} (V_1^*(s_1) - V_1^{\pi^k}(s_1)) \leq \sum_{k=1}^{K} (V_1^k(s_1) - V_1^{\pi^k}(s_1)).
$$

Because the action taking by $\pi^k$ is equal to action taking at episode $k$, we will have

$$
V_h^k(s_h^k) - V_h^{\pi^k}(s_h^k) = (Q_h^k - Q_h^{\pi^k})(s_h^k, a_h^k),
$$

where $a_h^k = \pi_h^k(s_h^k)$. Then

$$
\begin{aligned}
V_h^k(s_h^k) - V_h^{\pi^k}(s_h^k) &= (Q_h^k - Q_h^{\pi^k})(s_h^k, a_h^k) \\
&= (\hat{\mathbb{P}}_h^k(V_{h+1}^k) - \mathbb{P}_h(V_{h+1}^{\pi^k}))(s_h^k, a_h^k) + b(N_h^k(s_h^k, a_h^k), \delta) \\
&= (\hat{\mathbb{P}}_h^k - \mathbb{P}_h)(V_{h+1}^*)(s_h^k, a_h^k) + (\hat{\mathbb{P}}_h^k - \mathbb{P}_h)(V_{h+1}^k - V_{h+1}^*)(s_h^k, a_h^k) \\
&\quad + P_h(V_{h+1}^k - V_{h+1}^{\pi^k}))(s_h^k, a_h^k) + b(N_h^k(s_h^k, a_h^k), \delta).
\end{aligned}
\tag{8}
$$

First, by Chernoff-Hoeffding's bound, $(\hat{\mathbb{P}}_h^k - \mathbb{P}_h)(V_{h+1}^*)(s_h^k, a_h^k) \leq b(N_h^k(s_h^k, a_h^k), \delta)$. Second, by the standard analysis of UCBVI (Theorem 1 in UCBVI), under the event $\mathcal{E}_2$, we will have

$$
\begin{aligned}
&(\hat{\mathbb{P}}_h^k - \mathbb{P}_h)(V_{h+1}^k - V_{h+1}^*)(s_h^k, a_h^k) \\
&\leq \sum_{s' \in \mathcal{S}} \left( \sqrt{\frac{2\mathbb{P}_h(s' \mid s_h^k, a_h^k)\iota}{N_h^k(s_h^k, a_h^k))}} + \frac{2\iota}{3N_h^k(s_h^k, a_h^k)} \right) (V_{h+1}^k - V_{h+1}^*)(s') \\
&\leq \sum_{s' \in \mathcal{S}} \left( \frac{\mathbb{P}_h(s' \mid s_h^k, a_h^k)}{H} + \frac{H\iota}{2N_h^k(s_h^k, a_h^k)} + \frac{2\iota}{3N_h^k(s_h^k, a_h^k)} \right) (V_{h+1}^k - V_{h+1}^*)(s') \\
&\leq \frac{1}{H}\mathbb{P}_h(V_{h+1}^k - V_{h+1}^*)(s_h^k, a_h^k) + \frac{2H^2 S\iota}{N_h^k(s_h^k, a_h^k)} \\
&\leq \frac{1}{H}\mathbb{P}_h(V_{h+1}^k - V_{h+1}^{\pi^k})(s_h^k, a_h^k) + \frac{2H^2 S\iota}{N_h^k(s_h^k, a_h^k)},
\end{aligned}
$$

where the first inequality holds under the event $\mathcal{E}_2$, and the second inequality is derived by $\sqrt{ab} \leq \frac{a+b}{2}$, and the last inequality is because $V_{h+1}^{\pi^k} \leq V_{h+1}^*$. Thus combining with Eq. (8), we can get

$$
V_h^k(s_h^k) - V_h^{\pi^k}(s_h^k) \leq \left(1 + \frac{1}{H}\right) \mathbb{P}_h(V_{h+1}^k - V_{h+1}^{\pi^k})(s_h^k, a_h^k) + 2b(N_h^k(s_h^k, a_h^k), \delta) + \frac{2H^2 S\iota}{N_h^k(s_h^k, a_h^k)}.
$$

Denote

$$
\begin{aligned}
\alpha_h^k &= \mathbb{P}_h(V_{h+1}^k - V_{h+1}^{\pi^k})(s_h^k, a_h^k) - (V_{h+1}^k - V_{h+1}^{\pi^k})(s_h^K, a_h^k). \\
\beta_h^k &= \frac{H^2 S\iota}{N_h^k(s_h^k, a_h^k)}. \\
\gamma_h^k &= b(N_h^k(s_h^k, a_h^k), \delta).
\end{aligned}
$$

we have

$$V_h^k(s_h^k) - V_h^{\pi^k}(s_h^k) \le \left(1 + \frac{1}{H}\right)(V_{h+1}^k - V_{h+1}^{\pi^k})(s_{h+1}^k) + \left(1 + \frac{1}{H}\right)\alpha_h^k + 2\beta_h^k + 2\gamma_h^k$$

$$\le \left(1 + \frac{1}{H}\right)(V_{h+1}^k - V_{h+1}^{\pi^k})(s_{h+1}^k) + 2\alpha_h^k + 2\beta_h^k + 2\gamma_h^k,$$

then by recursion,

$$V_1^k(s_1) - V_1^{\pi^k}(s_1) \le \sum_{h=1}^H \left(1 + \frac{1}{H}\right)^H (2\alpha_h^k + 2\beta_h^k + 2\gamma_h^k)$$

$$\le e \cdot \sum_{h=1}^H (2\alpha_h^k + 2\beta_h^k + 2\gamma_h^k).$$

Under the event $\mathcal{E}_3$ with Bernstein's inequality, $\sum_{k=1}^K \sum_{h=1}^H \alpha_h^k \le \sqrt{\frac{H^3 K}{2} \log(1/\delta)}$, and

$$\sum_{k=1}^K \sum_{h=1}^H \beta_h^k = \sum_{k=1}^K \sum_{h=1}^H \frac{H^2 S\iota}{N_h^k(s_h^k, a_h^k)} \le H^3 S^2 A\iota \log(HK). \tag{9}$$

Also, for $\gamma_h^k$, we can get

$$\sum_{k=1}^K \sum_{h=1}^H \gamma_h^k = \sum_{k=1}^K \sum_{h=1}^H 7H \sqrt{\frac{\iota}{N_h^k(s_h^k, a_h^k)}}$$

$$\le 7H\sqrt{\iota} \sum_{(s,a)\in\mathcal{S}\times\mathcal{A}} \sum_{i=1}^{N^k(s,a)} \sqrt{1/i}$$

$$\le 14H\sqrt{\iota} \sum_{(s,a)\in\mathcal{S}\times\mathcal{A}} \sqrt{N^k(s,a)}$$

$$\le 14H\sqrt{\iota SAT}.$$

Then the regret will be bounded by $O(\sqrt{H^2 SAT})$.

**Violation** Now we turn to consider the violation during the process.

First, if the agent follows the $\pi^k$ at episode $k$, and $s_{h+1}^k \in \Delta_h^K(s_h^k, a_h^k)$ for all step $h \in [H-1]$, from the definition of $\{\mathcal{U}_h\}_{1\le h\le H+1}$ and $\Delta_h^K(s_h^k, a_h^k)$, all states $s_1^k, s_2^k, \cdots, s_H^k \notin \mathcal{U}_H$ are safe. Since $\mathcal{U}_H = \{s \mid \bar{c}(s) > \tau\}$, for each state $s \in \{s_1^k, s_2^k, \cdots, s_H^k\}$, we can have

$$c(s) - \tau \le \bar{c}^k(s) + 2\sqrt{\frac{2}{N^k(s)} \log(SK/\delta)} - \tau \le 2\sqrt{\frac{2}{N^k(s)} \log(SK/\delta)},$$

where the first inequality from the event $\mathcal{E}_1$ and the second inequality is because $s \notin \mathcal{U}_H^k$. Then

$$\sum_{h=1}^H (c(s_h^k) - \tau)_+ \le \sum_{h=1}^H 2\sqrt{\frac{2}{N^k(s_h^k)} \log(SK/\delta)}.$$

Then we consider the situation that $s_{h+1}^k \notin \Delta_h^K(s_h^k, a_h^k)$ for some step $h \in [H]$ at episode $k \in [K]$. In this particular case, we can only bound the violation on this episode by $H$ because the future state are not in control. Fortunately, in this situation we will add at least one new state $s_{h+1}^k$ to $\Delta_h^K(s_h^k, a_h^k)$, then $|\Delta_h^{k+1}(s_h^k, a_h^k)| \ge |\Delta_h^k(s_h^k, a_h^k)| + 1$, and the summation

$$\sum_{(h,s,a)\in[H]\times\mathcal{S}\times\mathcal{A}} |\Delta_h^{k+1}(s,a)| \ge \sum_{(h,s,a)\in[H]\times\mathcal{S}\times\mathcal{A}} |\Delta_h^k(s,a)| + 1.$$

Note that for any episode $k$, the summation has a trivial upper bound

$$\sum_{(h,s,a)\in[H]\times\mathcal{S}\times\mathcal{A}} |\Delta_h^k(s,a)| \le S^2 AH,$$

this situation will also appears for at most $S^2 AH$ episode and leads at most $S^2 AH^2$ extra violation. Hence the total violation can be upper bounded by

$$
\begin{aligned}
\sum_{k=1}^{K}\sum_{h=1}^{H}(c(s_h^k)-\tau)_+ &\le S^2 AH^2 + \sum_{k=1}^{K}\sum_{h=1}^{H}\left(2\sqrt{\frac{2}{N^k(s_h^k)}\log(SK/\delta)}\wedge 1\right) \\
&\le S^2 AH^2 + 2\sqrt{2\log(SK/\delta)}\sum_{k=1}^{K}\sum_{h=1}^{H}\left(\sqrt{\frac{1}{N^k(s_h^k)\vee 1}}\right) \\
&\le S^2 AH^2 + 2\sqrt{2\log(SK/\delta)}\sum_{s\in\mathcal{S}}\sum_{i=1}^{K}\left(\frac{N^{i+1}(s)-N^i(s)}{\sqrt{N^i(s)\vee 1}}\right) \qquad (10)\\
&\le S^2 AH^2 + 2\sqrt{8\log(SK/\delta)}\sum_{s\in\mathcal{S}}\sqrt{N^K(s)} \\
&\le S^2 AH^2 + 2\sqrt{8\log(SK/\delta)}\sqrt{ST} \\
&= \widetilde{\mathcal{O}}(S^2 AH^2 + \sqrt{ST})
\end{aligned}
$$

with probability at least $1-3\delta$. The inequality Eq. (10) is derived by Lemma 19 in Auer et al. (2008). By replacing $\delta$ to $\delta/3$ we can complete our proof.

Moreover, for the unsafe state set $\mathcal{U}$, recall that $d = \min_{s\in\mathcal{U}}(c(s)-\tau) > 0$. For the state $s_h^k$, if $s_h^k \notin \mathcal{U}_H^k$, we have

$$c(s_h^k) - \tau \le \bar{c}^k(s_h^k) + 2\sqrt{\frac{2}{N^k(s_h^k)}\log(SK/\delta)} - \tau \le 2\sqrt{\frac{2}{N^k(s_h^k)}\log(SK/\delta)}.$$

Thus we can get

$$N^k(s_h^k) \le \frac{8}{(c(s)-\tau)^2}\log(SK/\delta).$$

Now denote $\mathcal{K}$ be the set of episodes $k$ that $s_h^{k+1} \in \Delta_h^k(s_h^k,a_h^k)$. Now for each $s \notin \mathcal{U}$, denote $k_s = \max\{k : \exists\, h \in [H], s = s_h^k\}$, then our total violation can be bounded by

$$
\begin{aligned}
\sum_{k=1}^{K}\sum_{h=1}^{H}(c(s_h^k)-\tau)_+ &\le S^2 AH^2 + \sum_{k\in\mathcal{K}}\sum_{h=1}^{H}(c(s_h^k)-\tau)_+ \\
&\le S^2 AH^2 + \sum_{s\notin\mathcal{U}}N^{k_s}(s)(c(s)-\tau)_+ \\
&\le S^2 AH^2 + \sum_{s\notin\mathcal{U}}\frac{8}{(c(s)-\tau)_+^2}(c(s_h^k)-\tau)_+\log(SK/\delta) \\
&\le S^2 AH^2 + \frac{8S}{d}\log(SK/\delta).
\end{aligned}
$$

$\square$

## A.2 Proof of Theorem 6.2

First, apply the same argument in the Appendix A.1, the violation during the exploration phase can be bounded by

$$\sum_{k=1}^{K}\sum_{h=1}^{H}(c(s_h^k) - \tau)_+ \leq S^2 AH^2 + \sum_{k=1}^{K}\sum_{h=1}^{H}\sqrt{\frac{2}{N^k(s_h^k)}\log(SK/\delta)}$$

$$\leq S^2 AH^2 + \sqrt{2\log(SK/\delta)}\sum_{s\in\mathcal{S}}\sum_{i=1}^{N^K(s)}\sqrt{\frac{1}{i}}$$

$$\leq S^2 AH^2 + \sqrt{8\log(SK/\delta)}\sum_{s\in\mathcal{S}}\sqrt{N^K(s)}$$

$$= \widetilde{\mathcal{O}}(S^2 AH^2 + \sqrt{ST}).$$

Now we prove the Algorithm 2 will return $\varepsilon$-safe and $\varepsilon$-optimal policy. Lemma A.5 shows that $\overline{W}_h^k(s_1, \pi_1^{k+1}(s_1))$ bounds the estimation error for any reward function $r$.

**Lemma A.5.** *For any feasible policy $\pi \in \Pi^k$, step $h$, episode $k$ and reward function $r$, defining the estimation error as $\hat{e}_h^{k,\pi}(s,a;r) = |\hat{Q}_h^{k,\pi}(s,a;r) - Q_h^{k,\pi}(s,a;r)|$. Then, with probability at least $1 - \delta$, we have*

$$\hat{e}_h^{k,\pi}(s,a;r) \leq \overline{W}_h^k(s,a).$$

First, we prove the Lemma A.5. We provide an lemma to show a high probability event for concentration.

**Lemma A.6** (Kaufmann et al. (2021)). *Define the event as*

$$\mathcal{E}_4 = \left\{\forall k,h,s,a, KL(\hat{\mathbb{P}}_h^k(\cdot \mid (s,a)), \mathbb{P}_h^k(\cdot \mid (s,a)) \leq \frac{\gamma(N_h^k(s,a), \delta)}{N_h^k(s,a)}\right\},$$

*where $\beta(n,\delta) = 2\log(SAHK/\delta) + (S-1)\log(e(1 + n/(S-1)))$. Then $\Pr(\mathcal{E}_4^c) \leq \delta$.*

By Bellman equations,

$$\hat{Q}_h^{\pi}(s,a;r) - Q_h^{\pi}(s,a;r) = \sum_{s'}(\hat{\mathbb{P}}_h(s' \mid s,a) - \mathbb{P}_h(s' \mid s,a))Q_{h+1}^{\pi}(s', \pi(s');r)$$

$$+ \sum_{s'}\hat{\mathbb{P}}_h(s' \mid s,a)(\hat{Q}_{h+1}^{\pi}(s', \pi(s');r) - Q_{h+1}^{\pi}(s', \pi(s');r)).$$

By the Pinkser's inequality and event $\mathcal{E}_4$,

$$\hat{e}_h^{\pi}(s,a;r) \leq \sum_{s'}|\hat{\mathbb{P}}_h(s' \mid s,a) - \mathbb{P}_h(s' \mid s,a)|Q_{h+1}^{\pi}(s', \pi(s');r)$$

$$+ \sum_{s'}\hat{\mathbb{P}}_h(s' \mid s,a)|\hat{Q}_{h+1}^{\pi}(s', \pi(s');r) - Q_{h+1}^{\pi}(s', \pi(s');r)|$$

$$\leq H\|\hat{\mathbb{P}}_h(s' \mid s,a) - \mathbb{P}_h(s' \mid s,a)\|_1 + \sum_{s'}\hat{\mathbb{P}}_h(s' \mid s,a)\hat{e}_{h+1}^{\pi}(s', \pi_{h+1}(s');r)$$

$$\leq H\sqrt{\frac{\gamma(N_h(s,a), \delta)}{N_h(s,a)}} + \sum_{s'}\hat{\mathbb{P}}_h(s' \mid s,a)\hat{e}_{h+1}^{\pi}(s, \pi_{h+1}(s');r).$$

By induction, if $\hat{e}_{h+1}^{\pi}(s,a;r) \leq \overline{W}_{h+1}(s,a)$, for $s \notin \mathcal{U}_h^k, a \in A_h^{k,safe}(s)$, we can have

$$\hat{e}_h^{\pi}(s,a;r) \leq \min\left\{H, H\sqrt{\frac{\gamma(N_h(s,a), \delta)}{N_h(s,a)}} + \sum_{s'}\hat{\mathbb{P}}_h(s' \mid s,a)\hat{e}_{h+1}^{\pi}(s, \pi_{h+1}(s');r)\right\}$$

$$\leq \min\left\{H, H\sqrt{\frac{\gamma(N_h(s,a), \delta)}{N_h(s,a)}} + \sum_{s'}\hat{\mathbb{P}}_h(s' \mid s,a)\max_{b\in A_{h+1}^{k,safe}(s')}\hat{e}_{h+1}^{\pi}(s',b;r)\right\} \quad (11)$$

$$\leq \min\left\{H, H\sqrt{\frac{\gamma(N_h(s,a), \delta)}{N_h(s,a)}} + \sum_{s'}\hat{\mathbb{P}}_h(s' \mid s,a)\max_{b\in A_{h+1}^{k,safe}(s')}\overline{W}_{h+1}(s',b)\right\}$$

$$\leq \overline{W}_h(s,a).$$

The second inequality is because $\hat{\mathbb{P}}_h(s' \mid s, a) = 0$ for all $s' \in \mathcal{U}_{h+1}$, and then for $s' \notin \mathcal{U}_{h+1}, \pi_{h+1}(s') \in A_{h+1}^{k,safe}(s')$. The third inequality holds by induction hypothesis, and the last inequality holds by the definition of $\overline{W}_h(s, a)$.

Also, for other state-action pair $(s, a)$, we can get

$$\hat{e}_h^\pi(s, a; r) \leq \min \left\{ H, H\sqrt{\frac{\gamma(N_h(s,a), \delta)}{N_h(s,a)}} + \sum_{s'} \hat{\mathbb{P}}_h(s' \mid s, a)\hat{e}_{h+1}^\pi(s, \pi_{h+1}(s'); r) \right\}$$

$$\leq \min \left\{ H, H\sqrt{\frac{\gamma(N_h(s,a), \delta)}{N_h(s,a)}} + \sum_{s'} \hat{\mathbb{P}}_h(s' \mid s, a)\max_{b \in A} \hat{e}_{h+1}^\pi(s', b; r) \right\}$$

$$\leq \min \left\{ H, H\sqrt{\frac{\gamma(N_h(s,a), \delta)}{N_h(s,a)}} + \sum_{s'} \hat{\mathbb{P}}_h(s' \mid s, a)\max_{b \in A} \overline{W}_{h+1}(s', b) \right\}$$

$$\leq \overline{W}_h(s, a).$$

Hence the Lemma A.5 is true. Now we assume the algorithm terminates at episode $K + 1$, for any feasible policy $\pi \in \Pi^K$, if we define the value function in model $\hat{M}$ as $\hat{V}$, we will get

$$|V_1^\pi(s_1) - \hat{V}_1^\pi(s_1)| = |Q_1^\pi(s_1, \pi_1(s_1)) - \hat{Q}_1^\pi(s_1, \pi_1(s_1))|$$

$$\leq \overline{W}_h^K(s_1, \pi_1(s_1))$$

$$\leq \overline{W}_h^K(s_1, \pi_1^{K+1}(s_1))$$

$$\leq \varepsilon/2.$$

where the second inequality is because $\pi^{K+1}$ is the greedy policy with respect to $\overline{W}_h^K$ and $\pi$ is in the feasible policy set $\Pi^K$ at episode $K$. Then for any reward function $r$, denote the $\hat{\pi}^* \in \Pi^K$ to be the optimal policy in estimated MDP $\hat{M}$. Recall that $\pi^* \in \Pi^K \subseteq \Pi^*$, we will have

$$V_1^{\pi^*}(s_1) - V_1^{\hat{\pi}^*}(s_1) \leq \hat{V}_1^{\pi^*}(s_1) - V_1^{\hat{\pi}^*}(s_1) + \varepsilon/2$$

$$\leq \hat{V}_1^{\hat{\pi}^*}(s_1) - V_1^{\hat{\pi}^*} + \varepsilon/2$$

$$\leq \varepsilon.$$

Now we consider the expected violation. We will show that for all feasible policy $\pi \in \Pi^K$, the expected violation will be upper bounded by $2\overline{W}_1(s_1, \pi_1(s_1))$.

We first consider the event for feasible policy $\pi$

$$\mathcal{P}_\pi = \{\forall h \in [H-1], s_{h+1} \in \Delta_h^K(s_h, a_h) \mid s_h, a_h \sim \pi\}.$$

Assume that the event holds, then because $\pi \in \Pi^K$ is the feasible policy with respect to $\Delta_h^K(s, a)$, we have $s_h \notin \mathcal{U}_h$ for all $1 \leq h \leq H + 1$. Then

$$(c(s_h) - \tau)_+ \leq \sqrt{\frac{2}{N^K(s)} \log(SK/\delta)}.$$

and under the event $\mathcal{P}_\pi$

$$\mathbb{E}_\pi \left[ \sum_{h=1}^H (c(s_h) - \tau)_+ \right] \leq \mathbb{E}_\pi \left[ \sum_{h=1}^H \sqrt{\frac{2}{N^K(s_h)} \log\left(\frac{SK}{\delta}\right)} \right]. \tag{12}$$

Now define

$$\overline{F}_h^\pi(s, a)$$

$$= \min \left\{ H, \sqrt{\frac{2}{N(s_h)} \log\left(\frac{SK}{\delta}\right)} + H\sqrt{\frac{\gamma(N_h(s,a), \delta)}{N_h(s,a)}} + \sum_{s'} \hat{\mathbb{P}}_h(s' \mid s, a)F_{h+1}^\pi(s', \pi_{h+1}(s')) \right\}.$$

Since $\sqrt{\frac{2}{N(s_h)} \log\left(\frac{SK}{\delta}\right)} \leq H\sqrt{\frac{\gamma(N_h(s,a),\delta)}{N_h(s,a)}}$, we have $\overline{F}_h^\pi(s,a) \leq \overline{W}_h(s,a)$ by recursion. The proof is similar to A.5.

Also denote

$$F_h^\pi(s,a) = \mathbb{E}_\pi\left[\sum_{h'=h}^H \sqrt{\frac{2}{N^K(s_h)} \log\left(\frac{SK}{\delta}\right)} \,\Bigg|\, s_h = s, a_h = a\right].$$

Next we prove $\overline{F}_h^\pi(s,a) \geq F_h^\pi(s,a)$. To show this fact, we use the induction. When $h = H+1$, $F_{H+1}^\pi(s,a) = \overline{F}_{H+1}^\pi(s,a) = 0$. Now assume that $\overline{F}_{h+1}^\pi(s,a) \geq F_{h+1}^\pi(s,a)$, then

$$\overline{F}_h^\pi(s,a)$$

$$= \min\left\{H, \sqrt{\frac{2}{N(s)} \log\left(\frac{SK}{\delta}\right)} + H\sqrt{\frac{\gamma(N_h(s,a),\delta)}{N_h(s,a)}} + \sum_{s'} \hat{\mathbb{P}}_h(s' \mid s,a)\overline{F}_{h+1}^\pi(s',\pi_{h+1}(s'))\right\}$$

$$\geq \min\left\{H, \sqrt{\frac{2}{N(s)} \log\left(\frac{SK}{\delta}\right)} + H\sqrt{\frac{\gamma(N_h(s,a),\delta)}{N_h(s,a)}} + \sum_{s'} \hat{\mathbb{P}}_h(s' \mid s,a)F_{h+1}^\pi(s',\pi_{h+1}(s'))\right\}$$

$$\geq \min\left\{H, \sqrt{\frac{2}{N(s_h)} \log\left(\frac{SK}{\delta}\right)} + H\|\hat{\mathbb{P}}_h(s' \mid s,a) - \mathbb{P}_h(s' \mid s,a)\|_1\right.$$

$$\left. + \sum_{s'} \hat{\mathbb{P}}_h(s' \mid s,a)F_{h+1}^\pi(s',\pi_{h+1}(s'))\right\}$$

$$\geq \min\left\{H, \sqrt{\frac{2}{N(s_h)} \log\left(\frac{SK}{\delta}\right)} + \sum_{s'} \mathbb{P}_h(s' \mid s,a)F_{h+1}^\pi(s',\pi_{h+1}(s'))\right\}$$

$$= F_h^\pi(s,a).$$

Thus by induction, we know $F_1^\pi(s_1,\pi_1(s_1)) \leq \overline{F}_1^\pi(s_1,\pi_1(s_1)) \leq \overline{W}_1^K(s,\pi_1(s_1))$ because $\sqrt{\frac{2}{N(s)} \log\left(\frac{SK}{\delta}\right)} \leq H\sqrt{\frac{\gamma(N_h(s,a),\delta)}{N_h(s,a)}}$.

From now, we have proved that under the condition $\mathcal{P}_\pi$, the expected violation can be upper bounded by $\overline{W}_1(s,\pi_1(s_1))$. Next we state that $\Pr(\mathcal{P}_\pi^c)$ is small. Recall that $\mathcal{P}_\pi = \{\forall\, h \in [H-1], s_{h+1} \in \Delta_h^K(s_h,a_h) \mid \pi\}$, we define

$$G_h^\pi(s,a) = \Pr\{\exists\, h' \in [h,H-1], s_{h'+1} \notin \Delta_h^K(s_{h'},a_{h'}) \mid s_h = s, a_h = a\}$$

and

$$\overline{G}_h^\pi(s,a)$$

$$= \min\left\{H, \frac{SH\log(S^2HA/\delta)}{N_h(s,a)} + H\sqrt{\frac{\gamma(N_h(s,a),\delta)}{N_h(s,a)}} + \sum_{s'} \hat{\mathbb{P}}_h(s' \mid s,a)\overline{G}_{h+1}^\pi(s',\pi_{h+1}(s'))\right\}.$$

Then by recursion, we can easily show that $\overline{G}_h^\pi(s,a) \leq \overline{W}_h^K(s,a)$ by $\log(S^2HA/\delta) \leq \gamma(N_h(s,a),\delta)$.

**Lemma A.7.** *For all $h \in [H]$ and feasible policy $\pi$, we have $\overline{G}_h^\pi(s,a) \geq HG_h^\pi(s,a)$.*

*Proof.* Define $\widetilde{G}_h^\pi(s,a)$ as

$$\widetilde{G}_h^\pi(s,a) = \min\left\{H, \frac{SH\log(S^2HA/\delta)}{N_h(s,a)} + \sum_{s'} \mathbb{P}_h(s' \mid s,a)\widetilde{G}_{h+1}^\pi(s',\pi_{h+1}(s'))\right\}.$$

We will show $\overline{G}_h^\pi(s,a) \geq \widetilde{G}_h^\pi(s,a) \geq HG_h^\pi(s,a)$. We prove these two inequalities by induction. First, for $h = H+1$, these inequalities holds. Now assume that they hold for $h+1$, then

$$\overline{G}_h^\pi(s,a)$$

$$= \min\left\{H, \frac{SH\log(S^2HA/\delta)}{N_h(s,a)} + H\sqrt{\frac{\gamma(N_h(s,a),\delta)}{N_h(s,a)}} + \sum_{s'}\hat{\mathbb{P}}_h(s' \mid s,a)\overline{G}_{h+1}^\pi(s',\pi_{h+1}(s'))\right\}$$

$$\geq \min\left\{H, \frac{SH\log(S^2HA/\delta)}{N_h(s,a)} + H\sqrt{\frac{\gamma(N_h(s,a),\delta)}{N_h(s,a)}} + \sum_{s'}\hat{\mathbb{P}}_h(s' \mid s,a)\widetilde{G}_{h+1}^\pi(s',\pi_{h+1}(s'))\right\}$$

$$\geq \min\left\{H, \frac{SH\log(S^2HA/\delta)}{N_h(s,a)} + H\|\hat{\mathbb{P}}_h(s' \mid s,a) - \mathbb{P}_h(s' \mid s,a)\|_1\right.$$

$$\left. + \sum_{s'}\hat{\mathbb{P}}_h(s' \mid s,a)\widetilde{G}_{h+1}^\pi(s',\pi_{h+1}(s'))\right\}$$

$$\geq \min\left\{H, \frac{SH\log(S^2HA/\delta)}{N_h(s,a)} + \sum_{s'}\mathbb{P}_h(s' \mid s,a)\widetilde{G}_{h+1}^\pi(s',\pi_{h+1}(s'))\right\}$$

$$= \widetilde{G}_h^\pi(s,a).$$

Now to prove that $\widetilde{G}_h^\pi(s,a) \geq HG_h^\pi(s,a)$, we need the following lemma.

**Lemma A.8.** *Fixed a step $h \in [H]$ and state-action pair $(s,a) \in \mathcal{S} \times \mathcal{A}$, then with high probability at least $1 - \delta$, for any $s' \in \Delta_h(s,a)$ with $P(s' \mid s,a) \geq \frac{\log(S/\delta)}{N_h(s,a)}$, $s' \in \Delta_h^K(s,a)$.*

*Proof.* Since we sample $s'$ from $P_h(s' \mid s,a)$ for $N_h(s,a)$ times, if $s' \notin \Delta_h^K(s,a)$, then $s'$ are not sampled. The probability that $s'$ are not sampled will be $(1-p)^{N_h(s,a)}$, where $p = \mathbb{P}_h(s' \mid s,a)$. When $p \geq \frac{\log(S/\delta)}{N_h(s,a)}$, the probability will be at most

$$(1-p)^{N_h(s,a)} \leq \left(1 - \frac{\log(S/\delta)}{N_h(s,a)}\right)^{N_h(s,a)} \leq e^{-\log(S/\delta)} = \frac{\delta}{S}.$$

Taking the union bound for all possible next state $s'$, the lemma has been proved. $\square$

By the Lemma A.8, define the event as

$$\mathcal{E}_5 = \left\{\forall\, h,s,a \text{ and } s' \in \Delta_h(s,a) \text{ with } P(s' \mid s,a) \geq \frac{\log(S^2HA/\delta)}{N_h(s,a)} : s' \in \Delta_h^K(s,a)\right\}.$$

Then $\Pr(\mathcal{E}_5^c) \leq \delta$. Under the event $\mathcal{E}_5$, at step $h \in [H]$ and state-action pair $(s_h,a_h)$, we have

$$\mathbb{P}(s_{h+1} \notin \Delta_h^K(s_h,a_h)) \leq \frac{S\log(S^2HA/\delta)}{N_h(s_h,a_h)}. \tag{13}$$

This inequality is because $s_{h+1} \notin \Delta_h^K(s_h,a_h)$ only happens when $P(s' \mid s,a) \leq \frac{\log(S^2HA/\delta)}{N_h(s_h,a_h)}$ under the event $\mathcal{E}_5$. Now we first decompose the $G_h^\pi(s,a)$ as

$$H \cdot G_h^\pi(s,a) = \Pr\{\exists\, h' \in [h, H-1], s_{h'+1} \notin \Delta_h^K(s_{h'},a_{h'}) \mid s_h = s, a_h = a, \pi\}$$

$$\leq H\sum_{h'=h}^{H-1}\Pr\{s_{h'+1} \notin \hat{Delta}_{h'}^K(s_{h'},a_{h'}) \mid s_h = s, a_h = a, \pi\}$$

$$\leq \mathbb{E}_\pi\left[\sum_{h'=h}^{H-1}\frac{SH\log(S^2HA/\delta)}{N_{h'}(s_{h'},a_{h'})}\right].$$

The first inequality is to decompose the probability into the summation over step $h$, and the second inequality is derived by Eq. (13). Now denote $g_h^\pi(s,a) = \min\left\{H, \mathbb{E}_\pi\left[\sum_{h'=h}^{H-1} \frac{SH\log(S^2HA/\delta)}{N_{h'}(s_{h'},a_{h'})}\right]\right\}$, then we can easily have $g_h^\pi(s,a) \leq \widetilde{G}_h^\pi(s,a)$ by recursion. In fact, if

$$
\begin{aligned}
& g_h^\pi(s,a) \\
&= \min\left\{H, \mathbb{E}_\pi\left[\sum_{h'=h}^{H-1} \frac{SH\log(S^2HA/\delta)}{N_{h'}(s_{h'},a_{h'})}\ \bigg|\ (s_h,a_h)=(s,a)\right]\right\} \\
&= \min\left\{H, \frac{SH\log(S^2HA/\delta)}{N_h(s,a)} + \mathbb{E}_\pi\left[\sum_{h'=h+1}^{H-1} \frac{SH\log(S^2HA/\delta)}{N_{h'}(s_{h'},a_{h'})}\ \bigg|\ (s_h,a_h)=(s,a)\right]\right\} \\
&= \min\bigg\{H, \frac{SH\log(S^2HA/\delta)}{N_h(s,a)} \\
&\qquad\qquad + \min\left\{H, \mathbb{E}_\pi\left[\sum_{h'=h+1}^{H-1} \frac{SH\log(S^2HA/\delta)}{N_{h'}(s_{h'},a_{h'})}\ \bigg|\ (s_h,a_h)=(s,a)\right]\right\}\bigg\} \\
&= \min\left\{H, \frac{SH\log(S^2HA/\delta)}{N_h(s,a)} + \sum_{s'\in\mathcal{S}} \mathbb{P}_h(s'\mid s,a) g_{h+1}^\pi(s',\pi_{h+1}(s'))\right\}
\end{aligned}
$$

has the same recursive equation as $\widetilde{G}_h^\pi(s,a)$. Since $g_H^\pi(s,a) = 0 \leq \widetilde{G}_H^\pi(s,a)$, by induction we can get $g_h^\pi(s,a) \leq \widetilde{G}_h^\pi(s,a)$, and then

$$
H \cdot G_h^\pi(s,a) \leq g_h^\pi(s,a) \leq \widetilde{G}_h^\pi(s,a)
$$

holds directly. The Lemma A.7 has been proved.

Now we return to our main theorem. Consider the policy $\pi \in \Pi^K$ is one of feasible policy at episode $K$. For each possible $\mathscr{T}_H = \{s_1, a_1, \cdots, s_H, a_H\}$, denote $\Pr_\pi(\mathscr{T}_H)$ be the probability for generating $\mathscr{T}_H$ following policy $\pi$ and true transition kernel. If the event $s_{h+1} \notin \Delta_h^K(s_h,a_h)$ happens, we define the trajectory $\mathscr{T}_H$ are "unsafe". Denote the set of all "unsafe" trajectories are $\mathcal{U}$, we have

$$
\begin{aligned}
& \mathbb{E}_\pi\left[\sum_{h=1}^{H}(c(s_h)-\tau)_+\right] \\
&= \sum_{\mathscr{T}_H} \Pr_\pi(\mathscr{T}_H)\left[\sum_{h=1}^{H}(c(s_h)-\tau)_+\right] \\
&\leq \sum_{\mathscr{T}_H\in\mathcal{U}} \Pr_\pi(\mathscr{T}_H)\left[\sum_{h=1}^{H}(c(s_h)-\tau)_+\right] + \sum_{\mathscr{T}_H\notin\mathcal{U}} \Pr_\pi(\mathscr{T}_H)\left[\sum_{h=1}^{H}(c(s_h)-\tau)_+\right] \\
&\leq H\sum_{\mathscr{T}_H\in\mathcal{U}} \Pr_\pi(\mathscr{T}_H) + \sum_{\mathscr{T}_H\notin\mathcal{U}} \Pr_\pi(\mathscr{T}_H)\left[\sum_{h=1}^{H}\sqrt{\frac{2}{N^K(s_h)}\log(SK/\delta)}\right] \qquad (14) \\
&\leq H\Pr_\pi\{\mathcal{P}_\pi\} + \sum_{\mathscr{T}_H} \Pr_\pi(\mathscr{T}_H)\left[\sum_{h=1}^{H}\sqrt{\frac{2}{N^K(s_h)}\log(SK/\delta)}\right] \\
&\leq H \cdot G_1^\pi(s,a) + F_1^\pi(s,a) \qquad (15) \\
&\leq 2\overline{W}_1^K(s_1,\pi_1(s_1)) \\
&\leq 2\overline{W}_1^K(s_1,\pi_1^{K+1}(s_1)) \\
&\leq \varepsilon.
\end{aligned}
$$

The Eq. (14) is because $\mathscr{T}_H \in \mathcal{U}$ implies that the event $\mathcal{P}_\pi$ happens, and we can apply Eq. (12). The Eq. (15) is derived by the definition of $F_1^\pi(s,a)$ and $G_1^\pi(s,a)$. Until now, we have proved that if the algorithm terminates, all the requirements are satisfied. Now we turn to bound the sample complexity. We first assume that $K \geq SA$

Define the average $q_h^k = \sum_{(s,a)} \mathbb{P}_h^{\pi^{k+1}}(s,a)\overline{W}_h^k(s,a)$. Since the $\pi^{k+1}$ are greedy policy with respect to $\overline{W}_h^k(s,a)$, we have

$$\overline{W}_h^k(s,a)$$
$$= \min\left\{H, M(N_h^k(s,a),\delta) + \sum_{s'}\hat{\mathbb{P}}_h^k(s'\mid s,a)\overline{W}_{h+1}^k(s',\pi_{h+1}^{t+1}(s'))\right\}$$
$$\leq \min\left\{H, M(N_h^k(s,a),\delta) + \sum_{s'}\mathbb{P}_h^k(s'\mid s,a)\overline{W}_{h+1}^k(s',\pi_{h+1}^{t+1}(s'))\right.$$
$$\left. +H\|\hat{\mathbb{P}}_h^k(s'\mid s,a) - \mathbb{P}_h^k(s'\mid s,a)\|_1\right\}$$
$$\leq M'(N_h^k(s,a),\delta) + \sum_{s'}\mathbb{P}_h^k(s'\mid s,a)\overline{W}_{h+1}^k(s',\pi_{h+1}^{t+1}(s'))$$

where $M'(n,\delta) = 3H\left(\sqrt{\frac{2\beta(n,\delta)}{n}} \wedge 1\right) + SH\left(\frac{\beta(n,\delta)}{n} \wedge 1\right)$. Now

$$q_h^k = \sum_{(s,a)}\mathbb{P}_h^{\pi^{k+1}}(s,a)\overline{W}_h^k(s,a)$$
$$\leq \sum_{(s,a)}\mathbb{P}_h^{\pi^{k+1}}(s,a)M'(N_h^k(s,a),\delta) + \sum_{(s',s,a)}\mathbb{P}_h^{\pi^{k+1}}(s,a)\mathbb{P}_h^k(s'\mid s,a)\overline{W}_{h+1}^k(s',\pi_{h+1}^{k+1}(s'))$$
$$= \sum_{(s,a)}\mathbb{P}_h^{\pi^{k+1}}(s,a)M'(N_h^k(s,a),\delta) + q_{h+1}^k.$$

Thus

$$\varepsilon/2 < q_1^k \leq \sum_{h=1}^{H}\sum_{(s,a)}\mathbb{P}_h^{\pi^{k+1}}(s,a)M'(N_h^k(s,a),\delta), \forall\, k \in [K-1],$$

where the first equality is because $q_1^k = \overline{W}_h^k(s_1,\pi^{k+1}(s_1)) > \varepsilon/2$. Sum over $k \in [K-1]$, we can get

$$K\varepsilon/2 \leq \sum_{k=0}^{K-1}\sum_{h=1}^{H}\sum_{(s,a)}\mathbb{P}_h^{\pi^{k+1}}(s,a)M'(N_h^k(s,a),\delta).$$

Define the pseudo-count as $\overline{N}_h^k(s,a) = \sum_{i=1}^{k}\mathbb{P}_h^{\pi^i}(s,a)$ and the event about the pseudo-count like Kaufmann et al. (2021):

$$\mathcal{E}_{cnt} = \left\{\forall\, k,h,s,a : N_h^k(s,a) \geq \frac{1}{2}\overline{N}_h^k(s,a) - \log(SAH/\delta)\right\}.$$

Then $\Pr\{\mathcal{E}_{cnt}\} \leq \delta$. The proof is provided in Kaufmann et al. (2021). Also, the Lemma 7 in Kaufmann et al. (2021) show that we can change $N_h^k(s,a)$ to $\overline{N}_h^k(s,a)$ up to a constant.

**Lemma A.9** (Kaufmann et al. (2021)). *Under the event $\mathcal{E}_{cnt}$, for any $s \in \mathcal{S}, a \in \mathcal{A}$ and $k$, if $\beta(x,\delta)/x$ is non-increasing for $x \geq 1$ and $\beta(x,\delta)$ is increasing, we have*

$$\frac{\gamma(N_h^k(s,a),\delta)}{N_h^k(s,a)} \wedge 1 \leq \frac{4\gamma(\overline{N}_h^k(s,a),\delta)}{\overline{N}_h^k(s,a) \vee 1}.$$

The proof of Lemma A.9 can be found in Kaufmann et al. (2021). Then we can have

$$K\varepsilon/2 \leq \sum_{k=0}^{K-1}\sum_{h=1}^{H}\sum_{(s,a)} \mathbb{P}_h^{\pi^{k+1}}(s,a)M'(N_h^k(s,a),\delta) \tag{16}$$

$$\leq \underbrace{3H\sum_{h=1}^{H}\sum_{k=0}^{K-1}\sum_{(s,a)}(\overline{N}_h^{k+1}(s,a) - \overline{N}_h^k(s,a))\sqrt{\frac{2\gamma(N_h^k(s,a),\delta)}{N_h^k(s,a)}} \wedge 1}_{(a)}$$

$$+ \underbrace{SH\sum_{h=1}^{H}\sum_{k=0}^{K-1}\sum_{(s,a)}(\overline{N}_h^{k+1}(s,a) - \overline{N}_h^k(s,a))\left(\frac{\gamma(N_h^k(s,a),\delta)}{N_h^k(s,a)} \wedge 1\right)}_{(b)}. \tag{17}$$

For part $(a)$, we upper bound it by

$$(a) \leq 3H\sum_{h=1}^{H}\sum_{k=0}^{K-1}\sum_{(s,a)}(\overline{N}_h^{k+1}(s,a) - \overline{N}_h^k(s,a))\sqrt{\frac{8\gamma(\overline{N}_h^k(s,a),\delta)}{\overline{N}_h^k(s,a) \vee 1}}$$

$$\leq 3\sqrt{2}H\sqrt{\gamma(K,\delta)}\sum_{h=1}^{H}\sum_{k=0}^{K-1}\sum_{(s,a)}\frac{(\overline{N}_h^{k+1}(s,a) - \overline{N}_h^k(s,a))}{\sqrt{\overline{N}_h^k(s,a) \vee 1}} \tag{18}$$

$$\leq 3(2+\sqrt{2})H\sqrt{\gamma(K,\delta)}\sum_{h=1}^{H}\sum_{(s,a)}\sqrt{\overline{N}_h^K(s,a) \vee 1}$$

$$\leq 3(2+\sqrt{2})H\sqrt{\gamma(K,\delta)}\sum_{h=1}^{H}\sum_{(s,a)}(1 + \sqrt{\overline{N}_h^K(s,a)})$$

$$\leq 3(2+\sqrt{2})H^2\sqrt{\gamma(K,\delta)}(SA + \sqrt{SAK}) \tag{19}$$

$$\leq 6(2+\sqrt{2})H^2\sqrt{\gamma(K,\delta)}\sqrt{SAK},$$

where Eq. (18) is because Lemma 19 in Auer et al. (2008), Eq. (19) is derived by $\sum_{(s,a)}\overline{N}_h^K(s,a) = K$ and Cauchy's inequality, and the last inequality is because we assume $K \geq SA$.

Also for part $(b)$,

$$(b) \leq SH\sum_{h=1}^{H}\sum_{k=0}^{K-1}\sum_{(s,a)}(\overline{N}_h^{k+1}(s,a) - \overline{N}_h^k(s,a))\frac{4\gamma(\overline{N}_h^k(s,a),\delta)}{\overline{N}_h^k(s,a) \vee 1}$$

$$\leq SH\gamma(K,\delta)\sum_{h=1}^{H}\sum_{(s,a)}\sum_{k=0}^{K-1}\frac{(\overline{N}_h^{k+1}(s,a) - \overline{N}_h^k(s,a))}{\overline{N}_h^k(s,a) \vee 1}$$

$$\leq SH\gamma(K,\delta)\sum_{h=1}^{H}\sum_{(s,a)}4\log(\overline{N}_h^K(s,a)+1) \tag{20}$$

$$\leq 4S^2AH^2\gamma(K,\delta)\log(K+1),$$

where the Eq. (20) is because the Lemma 9 in Ménard et al. (2021). Now from the upper bound of $(a)$ and $(b)$, we can get

$$K\varepsilon/2 \leq 6(2+\sqrt{2})H^2\sqrt{\gamma(K,\delta)}\sqrt{SAK} + 4S^2AH^2\gamma(K,\delta)\log(K+1).$$

Recall that $\gamma(n,\delta) = 2\log(SAHK/\delta) + (S-1)\log(e(1+n/(S-1))) \geq 1$, by some technical transformation, performing Lemma C.1 we can get

$$K = \widetilde{\mathcal{O}}\left(\left(\frac{H^4SA}{\varepsilon^2} + \frac{S^2AH^2}{\varepsilon}\right)\left(\log\left(\frac{1}{\delta}\right) + S\right)\right),$$

under the event $\mathcal{E}_{cnt}, \mathcal{E}_1, \mathcal{E}_4, \mathcal{E}_5$. Hence the theorem holds with probability at least $1 - 4\delta$, and we can prove the theorem by replacing $\delta$ to $\delta/4$.

$\square$

### A.3 Proof of Theorem 5.1

*Proof.* We construct n MDP models with the same transition model but different safety cost. For each model, the states consist of a tree with degree $A$ and layer $H/3$. For each state inside the tree (not a leaf), the action $a_i$ will lead to the $i-$th branch of the next state. For leaf nodes, they are absorbing states, and we call them $s_1, s_2, \cdots, s_n$. Then $n = A^{H/3} = \frac{S(A-1)+1}{A} \geq \frac{S}{2}$.

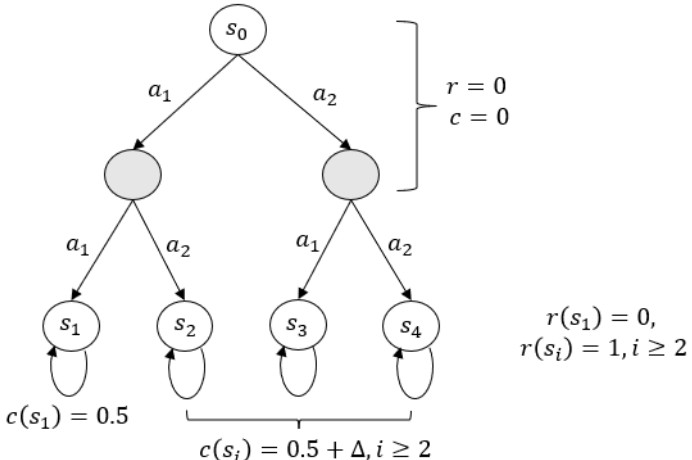

Figure 2: MDP instance $\tau_1$

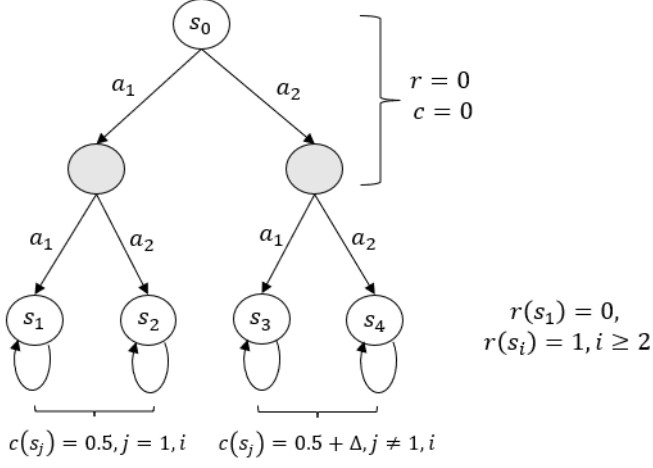

Figure 3: MDP instance $\tau_i (i = 2)$

Now we first define the instance $\tau_1$ as follows: For each state inside the tree, the safety cost and reward are all 0. For leaf node, the cost function for state $s_i$ are $c(s_i) = 1/2 + \Delta \mathbb{I}\{i \neq 1\}$. The reward function are $r(s_i) = \mathbb{I}\{i \neq 1\}$, which implies that only state $s_1$ has the reward 0 and other states $s_2, \cdots, s_n$ have reward 1. Now we define the instance $\tau_j (2 \leq j \leq n)$. For state inside the tree, the safety cost and reward are equal to the instance $\tau_1$. For leaf node, the cost function for state $s_i$ are $c(s_i) = 1/2 + \Delta \mathbb{I}\{i \neq 1, j\}$, $r(s_i)$ unchanged. Then for $\tau_j$, the state $s_1$ and $s_j$ (and initial state $s_0$)

are safe. Then the safe optimal policy is to choose $a_j$ at the first step. Thus the safe optimal reward is $H$. The MDP instances are shown in Figure 2 and Figure 3.

Denote $H' = \frac{2}{3}H$ are the rest of length when we arrive at the leaf states $s_1, \cdots, s_n$. Let $R_K(\pi, \tau)$ represents the regret till step episode $K$ for instance $\tau$ and policy $\pi$, and $C_K(\pi, \tau)$ represents the violation. For $\tau_1$, we can have $C_K(\pi, \tau_1) \geq \mathbb{P}_{\tau_1}(t_1(K) \leq K/2) \cdot \frac{KH'\Delta}{2}$, where $t_i(K)$ means the times to choose the path to $s_i$. Also, $R_K(\pi, \tau_i) \geq \mathbb{P}_{\tau_i}(t_1(K) > K/2)\frac{KH'}{2}$.

Applying Bertagnolle-Huber inequality, we have

$$\mathbb{P}_{\tau_1}(t_1(K) \leq K/2) + \mathbb{P}_{\tau_i}(t_1(K) > K/2) \geq \frac{1}{2}\exp\{-D(\mathbb{P}_{\tau_1}, \mathbb{P}_{\tau_i})\}.$$

We choose $i$ such that $i = \arg\min_i \mathbb{E}_{\tau_1}[t_i(K)]$, then by $\sum_{i \neq 1} \mathbb{E}_{\tau_1}[t_i(K)] \leq K$,

$$\mathbb{E}_{\tau_1}[t_i(K)] \leq \frac{K}{n-1}.$$

By the definition of $\mathbb{P}_{\tau_1}$ and $\mathbb{P}_{\tau_i}$,

$$\begin{aligned}
D(\mathbb{P}_{\tau_1}, \mathbb{P}_{\tau_i}) &= H'\mathbb{E}_{\tau_1}[t_i(K)]D(\mathcal{N}(1/2+\Delta, 1), \mathcal{N}(1/2, 1)) \\
&\leq H'\mathbb{E}_{\tau_1}[t_i(K)] \cdot 4\Delta^2 \\
&\leq \frac{4KH'\Delta^2}{n-1}.
\end{aligned}$$

Thus choose $\Delta = \sqrt{n-1/4KH'} \leq \frac{1}{2}$, (need $n \leq T$)

$$\mathbb{P}_{\tau_1}(t_1(K) \leq K/2) + \mathbb{P}_{\tau_i}(t_1(K) > K/2) \geq \frac{1}{2}\exp\{-D(\mathbb{P}_{\tau_1}, \mathbb{P}_{\tau_i})\} \geq \frac{1}{2e}.$$

If $\mathbb{E}_\pi(R_K(\pi, \tau)) \leq \frac{T}{24} = \frac{KH}{24} \leq \frac{KH'}{16}$ for all $\tau$, then

$$\mathbb{P}_{\tau_i}(t_1(K) > K/2) \leq \frac{2R_K(\pi, \tau_i)}{KH'} \leq \frac{1}{8}.$$

Thus

$$\mathbb{P}_{\tau_1}(t_1(K) \leq K/2) \geq \frac{1}{2e} - \frac{1}{8} := C$$

and

$$C_K(\pi, \tau_1) \geq \frac{CKH'\Delta}{2} = \frac{C\sqrt{(n-1)KH'}}{4} = \Omega(\sqrt{SHK}) = \Omega(\sqrt{SHK}),$$

where $n \geq \frac{S}{2}$. $\qquad\square$

## A.4  Proof of Theorem 5.2

*Proof.* Construct the MDP $\tau_1, \tau_2, \cdots, \tau_n$ as follows: For each instance, the states consist of a tree with degree $A-1$ and layer $H/3$. For each state inside the tree (not a leaf), the action $a_i, 1 \leq i \leq A-1$ will lead to the $i$-th branch of the next state. For the leaf nodes, denote them as $s_1, s_2, \cdots, s_n$, where $n = (A-1)^{H/3} = \frac{(S-3)(A-2)+1}{A-1} = \Omega(S)$ and these states will arrive at the final absorbing state $s_A$ and $s_B$ with reward $r(s_A) = 0$ and $r(s_B) = 1$ respectively. Also, the action $a_A$ will always lead the agent to the unique unsafe state $s_U$ with $r(s_U) = 0.5 + \Delta'$ and $c(s_U) = 1$ for some parameter $\Delta'$ in the states except two absorbing states. For all states except the unsafe state, the safety cost is 0.

Now for the instance $\tau_1$, the transition probability between the leaf states and absorbing states are

$$\begin{cases} p(s_A \mid s_1, a) = 0.5 - \Delta \\ p(s_B \mid s_1, a) = 0.5 + \Delta \\ p(s_A \mid s_i, a) = p(s_B \mid s_1, a) = 0.5 \quad i \geq 2 \end{cases}$$

For instance $\tau_j$, the transition probability are

$$\begin{cases} p(s_A \mid s_1, a) = 0.5 - \Delta \\ p(s_B \mid s_1, a) = 0.5 + \Delta \\ p(s_A \mid s_j, a) = 0.5 - 2\Delta \\ p(s_B \mid s_j, a) = 0.5 + 2\Delta \\ p(s_A \mid s_i, a) = p(s_B \mid s_1, a) = 0.5 \quad i \neq 1, j \end{cases}$$

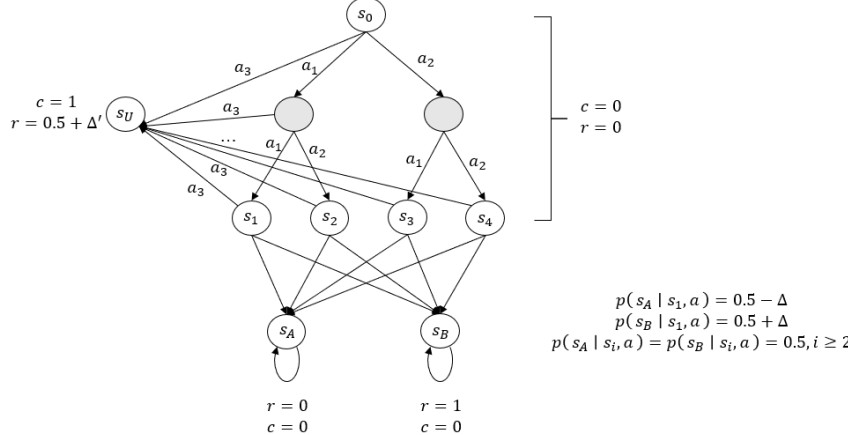

Figure 4: MDP instance $\tau_1$

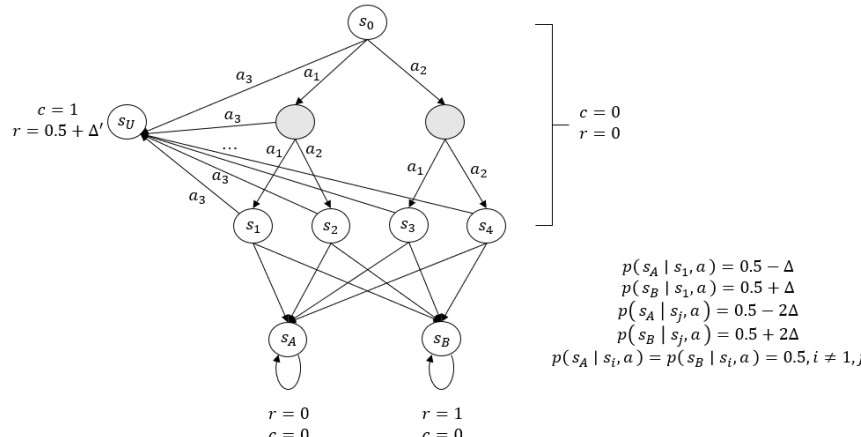

Figure 5: MDP instance $\tau_i$

Then if we do not consider the constraint, choosing $a_A$ and getting into the unsafe state $s_U$ is always the optimal action, then define $\text{gap}_{h,s,a} = V_h^*(s) - Q_h^*(s,a)$, then the minimal gap Simchowitz & Jamieson (2019) is

$$\text{gap}_{\min} = \min_{h,s,a}\{\text{gap}_{h,s,a} > 0 : \text{gap}_{h,s,a}\} \geq H'((0.5 + \Delta') - (0.5 + 2\Delta)) \geq H'(\Delta' - 2\Delta).$$

Thus the regret will be at most $O(\frac{S^2 A \cdot poly(H) \log T}{\Delta' - \Delta})$ by classical gap-dependent upper bound Simchowitz & Jamieson (2019). As we will show later, we will choose $\Delta = \Theta(T^{-1/2})$ and $\Delta' = \Theta(T^{\frac{\alpha-1}{2}})$, hence the regret will be

$$O\left(\frac{S^2 A \cdot poly(H)\log T}{\Delta' - \Delta}\right) = \tilde{\mathcal{O}}\left(T^{\frac{1-\alpha}{2}}\right).$$

Now if we consider the safe threshold $\tau = 0$, the violation will be the number of time to arrive the unsafe state. Now consider any algorithm, if it achieves $O(T^{1-\alpha})$ violation, they will arrive at unsafe state at most $O(T^{1-\alpha})$ times. Since denote $T_i(K), 1 \leq i \leq n+1$ as the number of times when agent arrive at state $s_i$ till episode $K$. Then $\mathbb{E}_\pi[T_{n+1}(K)] \leq CT^{1-\alpha}$. Note that each episode we will arrive at least one of $s_i, 1 \leq i \leq n+1$ once, we know $\sum_{i=1}^{n+1} T_i(K) \geq K$, and $\sum_{i=1}^{n} T_i(K) \geq K - C \cdot T^{1-\alpha} \geq K/2$ when $K \geq (2C)^{1/\alpha} H^{\frac{1-\alpha}{\alpha}}$.

Now similar to proof of Theorem 5.1, let $C_K(\pi, \tau)$ represents the violation with algorithm $\pi = \{\tau_1, \cdots, \pi_K\}$. Denote $H' = \frac{2}{3}H$. For $\tau_1$, we have

$$R_K(\pi, \tau_1) \geq \mathbb{P}_{\tau_1}(T_1(K) \leq K/4)\frac{KH'\Delta}{4} - (\Delta' - \Delta)\mathbb{E}_\pi[T_{n+1}(K)]$$
$$\geq \mathbb{P}_{\tau_1}(T_1(K) \leq K/4)\frac{KH'\Delta}{4} - (\Delta' - \Delta)CT^{1-\alpha}$$

$$R_K(\pi, \tau_i) \geq \mathbb{P}_{\tau_i}(T_1(K) > K/4)\frac{KH'\Delta}{4} - (\Delta' - 2\Delta)\mathbb{E}_\pi[T_{n+1}(K)]$$
$$\geq \mathbb{P}_{\tau_i}(T_1(K) > K/4)\frac{KH'\Delta}{4} - (\Delta' - 2\Delta)CT^{1-\alpha}$$

Also, we have

$$\mathbb{P}_{\tau_1}(T_1(K) \leq K/4) + \mathbb{P}_{\tau_i}(T_1(K) > K/4) \geq \frac{1}{2}\exp\{-D(\mathbb{P}_{\tau_1}, \mathbb{P}_{\tau_i})\}$$

Now WLOG we have $\mathbb{E}_{\tau_1}[T_i(K)] \leq \frac{K}{2(n-1)}$

$$D(\mathbb{P}_{\tau_1}, \mathbb{P}_{\tau_i}) \leq \mathbb{E}_{\tau_1}[T_i(K)]2\Delta^2 \leq \frac{K}{2(n-1)} \cdot 2\Delta^2 = \frac{K\Delta^2}{n-1}$$

Choose $\Delta = \sqrt{\frac{n-1}{K}}$, then

$$\mathbb{P}_{\tau_1}(T_1(K) \leq K/4) + \mathbb{P}_{\tau_i}(T_1(K) > K/4) \geq \frac{1}{2e}.$$

Then at least one of $\tau_1$ and $\tau_i$ (assume $\tau_1$) have

$$R_K(\pi, \tau_1) \geq \frac{H'\sqrt{K(n-1)}}{8e} - C(\Delta' - 2\Delta)T^{1-\alpha}$$

Choose $\Delta' = T^{\frac{\alpha-1}{2}}$. Then assume $T \geq 2^{\frac{2}{1-\alpha}}$ so that $c \leq \frac{1}{2}$.

$$R_K(\pi, \tau_1) \geq \frac{H'\sqrt{K(n-1)}}{8e} - CT^{(1-\alpha)/2} = \Omega(H\sqrt{SK}) = \Omega(\sqrt{HST}).$$

$\square$

# B  Safe Markov Games

## B.1  Definitions

In this section, we provide an extension for our general framework of Safe-RL-SW. We solve the safe zero-sum two-player Markov games Yu et al. (2021).

In zero-sum two-player Markov games, there is an agent and another adversary who wants to make the agent into the unsafe region and also minimize the reward. The goal of agent is to maximize the reward and avoid the unsafe region with the existence of adversary. To be more specific, define the action space for the agent and adversary are $\mathcal{A}$ and $\mathcal{B}$ respectively. In each episode the agent starts at an initial state $s_1$. At each step $h \in [H]$, the agent and adversary observe the state $s_h$ and taking action $a_h \in \mathcal{A}$ and $b_h \in \mathcal{B}$ from the strategy $\mu_h(\cdot \mid s)$ and $\upsilon_h(\cdot \mid s)$ simultaneously. Then the agent receive a reward $r_h(s, a, b)$ and the environment transitions to the next state $s_{h+1} \sim P(s_{h+1} \mid s, a, b)$ by the transition kernel $P$.

One motivation of this setting is that, when a robot is in a dangerous environment, the robot should learn a completely safe policy even with the disturbance from the environment. Like the single-player

MDP, we define the value function and Q-value function:

$$V_h^{\mu,\upsilon}(s) = \mathbb{E}_\pi\left[\sum_{h'=h}^{H} r_{h'}(s_{h'}, a_{h'}, b_{h'})\middle| s_h = s, \mu, \upsilon\right].$$

$$Q_h^{\mu,\upsilon}(s, a, b) = \mathbb{E}_\pi\left[\sum_{h'=h}^{H} r_{h'}(s_{h'}, a_{h'}, b_{h'})\middle| s_h = s, a_h = a, b_h = b, \mu, \upsilon\right].$$

The cost function is a bounded function defined on the states $c(s) \in [0, 1], s \in \mathcal{S}$, and the unsafe states are defined $\mathcal{U}_H = \{s \mid c(s) > \tau\}$ for some fixed threshold $\tau$. Similar to the Section 4, we define the possible next states for state $s$, step $h$, action pair $(a, b)$ as

$$\Delta_h(s, a, b) = \{s' \mid \mathbb{P}_h(s' \mid s, a, b) > 0\}.$$

Hence we can define the unsafe states for step $h$ recursively as:

$$\mathcal{U}_h = \mathcal{U}_{h+1} \cup \{s \mid \forall \in \mathcal{A}, \exists b \in \mathcal{B}, \Delta_h(s, a, b) \cap \mathcal{U}_{h+1} \neq \emptyset\}.$$

Then to keep safety, the agent cannot arrive at the state $s \in \mathcal{U}_h$ at the step $1, 2, \cdots, h$. The safe action for the agent $A_h^{safe}(s)$ should avoid the unsafe region no matter what action the adversary takes.

$$A_h^{safe}(s) = \{a \in \mathcal{A} \mid \forall b \in \mathcal{B}, \Delta_h(s, a, b) \cap \mathcal{U}_{h+1} = \emptyset\}.$$

Hence a feasible policy should keep the agent in safe state $s \notin \mathcal{U}_h$ at step $h$, and always chooses $a \in A_h^{safe}(s)$ to avoid the unsafe region regardless of the adversary. Analogous to the safe RL in Section 4, assume the the optimal Bellman equation can be written as

$$Q_h^*(s, a, b) = r(s, a) + \mathbb{P}_h(s' \mid s, a, b)V_{h+1}^*(s').$$
$$V_h^*(s) = \max_{a \in A_h^{safe}(s)} \min_{b \in \mathcal{B}} Q_h^*(s, a, b).$$

If we denote $r(s, a, b) = -\infty$ for unsafe states $s \in \mathcal{U}_H$, fixed an strategy $\upsilon$ for the adversary, the *best response* of the agent can be defined as $\arg\max_\mu V_1^{\mu,\upsilon}(s_1)$. Similarly, a *best response* of the adversary given agent's strategy $\mu$ can be defined as $\arg\min_\upsilon V_1^{\mu,\upsilon}(s_1)$. By the minimax equality, for any step $h \in [H]$ and $s \in \mathcal{S}$,

$$\max_\mu \min_\upsilon V_h^{\mu,\upsilon}(s) = \min_\upsilon \max_\mu V_h^{\mu,\upsilon}(s).$$

A policy pair $(\mu^*, \upsilon^*)$ satisfy that $V_h^{\mu^*,\upsilon^*}(s) = \max_\mu \min_\upsilon V_h^{\mu,\upsilon}(s)$ for all step $h \in [H]$ is called a *Nash Equilibrium*, and the value $V_h^{\mu^*,\upsilon^*}(s)$ is called the *minimax value*. It is known that minimax value is unique for Markov games, and the agent's goal is make the reward closer to the minimax value.

Assume the agent and adversary executes the policy $(\mu_1, \upsilon_1), (\mu_2, \upsilon_2), \cdots, (\mu_K, \upsilon_K)$ for episode $1, 2, \cdots, K$, the regret is defined as

$$R(K) = \sum_{k=1}^{K} V_1^{\mu^*,\upsilon^*}(s_1) - V_1^{\mu_k,\upsilon_k}(s_1).$$

Also, the actual state-wise violation is defined as

$$C(K) = \sum_{k=1}^{K}\sum_{h=1}^{H}(c(s_h^k) - \tau)_+.$$

## B.2 Algorithms and Results

The main algorithm is presented in Algorithm 3. The key idea is similar to the Algorithm 1. However, in each episode we need to calculate an optimal policy $\pi^k$ by a sub-procedure "Planning". This sub-procedure calculates the Nash Equilibrium for the estimated MDP. Note that there is always a mixed Nash equilibrium strategy $(\mu, \upsilon)$, and the ZERO-SUM-NASH function in "Planning" procedure calculates the Nash equilibrium for a Markov games.

**Algorithm 3** Safe Markov Games

1: Input: $\Delta_h^1(s, a, b) = \emptyset, N_h^1(s, a, b), N_h^1(s, a, b, s') = 0$.
2: **for** $k = 1, 2, \cdots, K$ **do**
3:    Receive the initial state $s_1$.
4:    Update the estimation $\hat{c}(s)$ based on history and get optimistic estimation $\bar{c}(s) = \hat{c}(s) - \beta(N^k(s), \delta)$ for each state $s$, where $\beta$ is the bouns function.
5:    Define $\mathcal{U}_H^k = \{s \mid \bar{c}(s) > \tau\}$.
6:    Based on $\Delta_h^K(s, a)$, calculate $\mathcal{U}_h^k = \mathcal{U}_{h+1}^k \cup \{s \mid \forall a \in A, \Delta_k(s, a, b) \cap \mathcal{U}_{h+1}^k \neq \emptyset\}$.
7:    $\hat{\mathbb{P}}_h(s' \mid s, a, b) = \frac{N_h^k(s, a, b, s')}{N_h^k(s, a, b)}$.
8:    $\pi_h^k = $Planning$(k, r, \{\mathcal{U}_h^k\}_{h \in [H]}, \Delta_h^K(s, a, b), N^k(s, a, b), \hat{\mathbb{P}})$.
9:    **for** $h = 1, 2, \cdots, H$ **do**
10:       Take action $a_h^k$ such that $(a_h^k, b) \sim \pi_h^k(\cdot, \cdot \mid s)$ and arrive state $s_{h+1}^k \sim P(s \mid s_h^k, a_h^k, b_h^k)$, where $b_h^k$ is the action of the adversary.
11:       $\Delta_h^{k+1}(s_h^k, a_h^k, b_h^k) = \Delta_h^k(s_h^k, a_h^k, b_h^k) \cup \{s_{h+1}^k\}$.
12:    **end for**
13:    Update $N_h^{k+1}(s, a, b), N_h^{k+1}(s, a, b, s'), \Delta_h^{k+1}(s, a, b)$ for all $(s, a, b)$.
14: **end for**

---

**Algorithm 4** Planning$(k, r, \{\mathcal{U}_h\}_{h \in [H]}, \Delta(s, a, b), N, \hat{\mathbb{P}})$

1: Input $k, r, \{\mathcal{U}_h\}, \Delta(s, a, b), N, \hat{\mathbb{P}}$.
2: **for** $h = H, H - 1, \cdots, 1$ **do**
3:    **for** $(s, a, b) \in \mathcal{S} \times \mathcal{A} \times \mathcal{B}$ **do**
4:       $\beta(N_h^k(s, a, b), \delta) \leftarrow 7H\sqrt{\log(5SABT/\delta)/N_h^k(s, a, b)}$.
5:       $Q_h^k(s, a, b) = \min\{(r_h + \sum_{s'} \hat{\mathbb{P}}_h^k(s' \mid s, a, b)V_{h+1}(s')) + \beta(N_h^k(s, a, b), \delta), H\}$.
6:    **end for**
7:    **for** $s \in S$ **do**
8:       **for** $a \in A$ **do**
9:          **if** $\exists b, \Delta_h(s, a, b) \cap \mathcal{U}_{h+1} \neq \emptyset$ **then**
10:             $(Q_h^k)'(s, a, \cdot) = -\infty$.
11:          **else**
12:             $(Q_h^k)'(s, a, \cdot) = Q_h^k(s, a, \cdot), a \in A_h^k(s)$.
13:          **end if**
14:       **end for**
15:       $\mu_h^k, \upsilon_h^k = $ZERO-SUM-NASH$((Q_h^k)'(s, \cdot, \cdot))$.
16:       $V_h^k(s) = \mathbb{E}_{a \sim \mu_h^k(\cdot|s), b \sim \upsilon_h^k(\cdot|s)}(a, b)Q_h(s, a, b)$.
17:    **end for**
18: **end for**

---

In the sub-procedure "Planning", we calculate the $(Q_h^k)'(s, a, \cdot) = -\infty$ if there exists at least one adversary action $b$ such that $\Delta(s, a, b) \cap \mathcal{U}_{h+1} \neq \emptyset$. In fact, the agent cannot take these actions at this time, hence by setting $(Q_h^k)'(s, a, \cdot)$ for these actions $a$, the support of Nash Equilibrium policy $\mu_h^k$ will not contains $a$. Otherwise, the minimax value is $-\infty$, and it means that $a$ is now in $\mathcal{U}_h^k$, then at this time the agent can chooes the action arbitrarily

Our main result are presented in the following theorem, which shows that our algorithm achieves both $O(\sqrt{T})$ regret and $O(\sqrt{ST})$ state-wise violation.

**Theorem B.1.** *With probability at least $1 - \delta$, the Algorithm 3 have regret and state-wise violation*

$$R(T) = O(H^3\sqrt{SABT})$$
$$C(T) = O(\sqrt{ST} + S^2ABH^2),$$

*where $A = |\mathcal{A}|$, $B = |\mathcal{B}|$ and $S = |\mathcal{S}|$.*

## B.3 Proofs

*Proof.* We first prove that, with probability at least $1 - \delta$, $Q_h^k(s, a, b) \geq Q_h^*(s, a, b)$ and $V_h^k(s) \geq V_h^*(s)$ for any episode $k \in [K]$. We prove this fact by induction. Denote $\mathbb{D}_{\mu \times \upsilon} Q(s) = \mathbb{E}_{a \sim \mu(\cdot|s), b \sim \upsilon(\cdot|s)} Q(s, a, b)$. The statement holds for $h = H + 1$. Suppose the bounds hold for $Q-$values in the step $h + 1$, now we consider step $h$.

$$
\begin{aligned}
V_{h+1}^k(s) &= \mathbb{D}_{\pi_h^k} Q_{h+1}^k(s) \\
&= \max_{supp(\mu) \subseteq A_h^{k,safe}(s)} \mathbb{D}_{\mu \times \upsilon_h^k} Q_{h+1}^k(s) \\
&\geq \max_{\mu \in supp(\mu) \subseteq A_h^k(s)} \mathbb{D}_{\mu \times \upsilon_h^k} Q_{h+1}^*(s) \\
&\geq \max_{\mu \in supp(\mu) \subseteq A_h^*(s)} \mathbb{D}_{\mu \times \upsilon_h^k} Q_{h+1}^*(s) \\
&\geq \min_\upsilon \max_{\mu \in supp(\mu) \subseteq A_h^*(s)} \mathbb{D}_{\mu \times \upsilon} Q_{h+1}^*(s) \\
&= V_{h+1}^*(s),
\end{aligned}
$$

where $A_h^{k,safe}(s)$ is all safe action for max-player at episode $k$ and step $h$ it estimated, $A_h^*(s)$ is the true safe action, and $supp(\mu) = \{a \in \mathcal{A} \mid \mu(a) > 0\}$ From our algorithm, we know $A_h(s) \subseteq A_h^k(s)$ for all time if the confidence event always holds.

Then

$$
\begin{aligned}
Q_h^k(s, a, b) - Q_h^*(s, a, b) &= (\hat{\mathbb{P}}_h^k V_{h+1}^k - \mathbb{P}_h V_{h+1}^* + \beta_h^k)(s, a, b) \\
&\geq (\hat{\mathbb{P}}_h^k - \mathbb{P}_h)(V_{h+1}^*)(s, a, b) + \beta(N_h^k(s, a, b), \delta) \\
&\geq 0.
\end{aligned}
$$

The last inequality is because of the Chernoff-Hoeffding's inequality. Thus define event

$$
\mathcal{E}_6 = \left\{ \forall k, h, s, a, b, \ Q_h^k(s, a, b) \geq Q_h^*(s, a, b), V_h^k(s) \geq V_h^*(s) \right\}.
$$

Then $\Pr\{\mathcal{E}_6^c\} \leq \delta$.

Now we start to bound the regret.

$$
\begin{aligned}
R(K) &= \sum_{k=1}^K V_1^{\mu^*, \upsilon^*}(s_1) - V_1^{\mu_k, \upsilon_k}(s_1) \\
&\leq \sum_{k=1}^K V_1^k(s_1) - V_1^{\mu_k, \upsilon_k}(s_1).
\end{aligned}
$$

Now we first assume $\upsilon_k$ is the best response of $\mu_k$. Otherwise our regret can be larger. We calculate the regret by

$$
\begin{aligned}
&V_h^k(s_h^k) - V_h^{\mu_k, \upsilon_k}(s_h^k) \\
&= \mathbb{D}_{\pi^k}(Q_h^k)(s_h^k) - \mathbb{D}_{\mu_k \times \upsilon_k} Q_h^{\mu_k \times \upsilon_k}(s_h^k) \\
&\leq \mathbb{D}_{\mu_k \times \upsilon_k}(Q_h^k)(s_h^k) - \mathbb{D}_{\mu_k \times \upsilon_k} Q_h^{\mu_k \times \upsilon_k}(s_h^k) \\
&= \mathbb{D}_{\mu_k \times \upsilon_k}(Q_h^k - Q_h^{\mu_k \times \upsilon_k})(s_h^k) \\
&= (Q_h^k - Q_h^{\mu_k \times \upsilon_k})(s_h^k, a_h^k, b_h^k) + \alpha_h^k \\
&= (\hat{\mathbb{P}}_h - \mathbb{P}_h) V_{h+1}^k(s_h^k, a_h^k, b_h^k) + \mathbb{P}_h \left( V_{h+1}^k - V_{h+1}^{\mu_k \times \upsilon_k} \right)(s_h^k, a_h^k, b_h^k) + \beta(N_h^k(s_h^k, a_h^k, b_h^k), \delta) + \alpha_h^k \\
&\leq \left( V_{h+1}^k - V_{h+1}^{\mu_k \times \upsilon_k} \right)(s_{h+1}^k) + \alpha_h^k + \gamma_h^k + 2\beta(N_h^k(s_h^k, a_h^k, b_h^k), \delta) \\
&\qquad\qquad\qquad\qquad\qquad + (\hat{\mathbb{P}}_h - \mathbb{P}_h)(V_{h+1}^k - V_{h+1}^*)(s_h^k, a_h^k, b_h^k),
\end{aligned}
$$

where

$$
\alpha_h^k = \mathbb{P}_h \left( V_{h+1}^k - V_{h+1}^{\mu_k \times \upsilon_k} \right)(s_h^k, a_h^k, b_h^k) - \left( V_{h+1}^k - V_{h+1}^{\mu_k \times \upsilon_k} \right)(s_{h+1}^k),
$$

$$\gamma_h^k = \left(\mathbb{D}_{\mu_k \times \upsilon_k}(Q_h^k - Q_h^{\mu_k \times \upsilon_k})(s_h^k) - (Q_h^k - Q_h^{\mu_k \times \upsilon_k})(s_h^k)\right).$$

The last inequality is because $(\hat{\mathbb{P}}_h - \mathbb{P}_h)V_{h+1}^*(s_h^k, a_h^k, b_h^k) \le \beta(N_h^k(s_h^k, a_h^k, b_h^k), \delta)$ by Chernoff-Hoeffding's inequality.

Now by the similar analysis in the previous section, with probability at least $1 - \delta$, the event

$$\mathcal{E}_7 = \Bigg\{ \forall s, a, b, s', k, h, |\hat{\mathbb{P}}_h(s' \mid s, a, b) - \mathbb{P}_h(s' \mid s, a, b)|$$
$$\le \sqrt{\frac{2\mathbb{P}_h(s' \mid s, a, b)}{N_h^k(s, a)} \log\left(\frac{2S^2 ABHK}{\delta}\right)} + \frac{2}{3N_h^k(s, a)} \log\left(\frac{2S^2 ABHK}{\delta}\right) \Bigg\}.$$

holds with probability at least $1 - \delta$ by Bernstein's inequality. Then for $\iota = \log\left(\frac{2S^2 ABHK}{\delta}\right)$

$$(\hat{\mathbb{P}}_h - \mathbb{P}_h)(V_{h+1}^k - V_{h+1}^*)(s_h^k, a_h^k, b_h^k)$$
$$\le \sum_{s'} \left(\sqrt{\frac{2\mathbb{P}_h(s' \mid s, a, b)}{N_h^k(s, a)}\iota} + \frac{2}{3N_h^k(s, a)}\iota\right)(V_{h+1}^k - V_{h+1}^*)(s')$$
$$\le \sum_{s'} \left(\frac{\mathbb{P}_h(s' \mid s_h^k, a_h^k)}{H} + \frac{H\iota}{2N_h^k(s_h^k, a_h^k)} + \frac{2\iota}{3N_h^k(s_h^k, a_h^k)}\right)(V_{h+1}^k - V_{h+1}^*)(s')$$
$$\le \frac{1}{H}\mathbb{P}_h(V_{h+1}^k - V_{h+1}^*)(s_h^k, a_h^k) + \frac{2H^2 S\iota}{N_h^k(s_h^k, a_h^k)}$$
$$\le \frac{1}{H}\mathbb{P}_h(V_{h+1}^k - V_{h+1}^{\mu_k \times \upsilon_k})(s_h^k, a_h^k) + \frac{2H^2 S\iota}{N_h^k(s_h^k, a_h^k)}.$$

The last inequality is because we assume $\upsilon_k$ is the best response of $\mu_k$ and then $V_{h+1}^*(s_h^k, a_h^k) \ge V_{h+1}^{\mu_k \times \upsilon_k}(s_h^k, a_h^k)$. Now we can get

$$V_h^k(s_h^k) - V_h^{\mu_k \times \upsilon_k}(s_h^k)$$
$$\le \left(1 + \frac{1}{H}\right)(V_{h+1}^k(s_h^k) - V_{h+1}^{\mu_k \times \upsilon_k}(s_h^k)) + 2\alpha_h^k + 2\gamma_h^k + 2\beta(N_h^k(s_h^k, a_h^k, b_h^k), \delta) + \frac{2H^2 S\iota}{N_h^k(s_h^k, a_h^k)}$$

and then

$$V_1^k(s_1) - V_1^{\mu_k \times \upsilon_k}(s_h^k) \le \sum_{h=1}^H \left(1 + \frac{1}{H}\right)^H (2\alpha_h^k + 2\gamma_h^k + 2\beta(N_h^k(s_h^k, a_h^k, b_h^k), \delta)) + \frac{2H^2 S\iota}{N_h^k(s_h^k, a_h^k)}$$
$$\le \sum_{h=1}^H e \cdot (2\alpha_h^k + 2\gamma_h^k + 2\beta(N_h^k(s_h^k, a_h^k, b_h^k), \delta)) + \frac{2H^2 S\iota}{N_h^k(s_h^k, a_h^k)}.$$

By Azuma-hoeffding's inequality, with probability at least $1 - 2\delta$, $\sum_{k=1}^K \sum_{h=1}^H \alpha_h^k \le O(\sqrt{H^3 K})$, $\sum_{k=1}^K \sum_{h=1}^H \gamma_h^k = O(\sqrt{H^3 K})$ and

$$\sum_{k=1}^K \sum_{h=1}^H \frac{2H^2 S\iota}{N_h^k(s_h^k, a_h^k)} = O(\log K).$$

Note that

$$\sum_{k=1}^K \sum_{h=1}^H \beta(N_h^k(s_h^k, a_h^k, b_h^k), \delta) = \sum_{k=1}^K \sum_{h=1}^H 7H\sqrt{\frac{\iota}{N_h^k(s_h^k, a_h^k, b_h^k)}}$$
$$\le O\left(H\iota \sum_{(h,s,a,b) \in [H] \times \mathcal{S} \times \mathcal{A} \times \mathcal{B}} \sqrt{N_h^k(s_h^k, a_h^k, b_h^k)}\right)$$
$$\le \tilde{O}(\sqrt{H^3 SABT}).$$

For the violation, the argument is similar to the Theorem 4.2. During the learning process, the $s_{h+1}^k \notin \Delta_h^K(s_h^k, a_h^k, b_h^k)$ can appear at most $S^2ABH$ times because each time the summation

$$\sum_{(h,s,a,b)\in[H]\times\mathcal{S}\times\mathcal{A}\times\mathcal{B}} |\Delta_h^K(s,a,b)|$$

will increase at least 1, and it has a upper bound $S^2ABH$. Thus it will lead to at most $S^2ABH^2$ regret. For other situations, the total violation can be upper bounded by $\widetilde{\mathcal{O}}(\sqrt{ST})$. Thus the final violation bound is $\tilde{O}(S^2ABH^2 + \sqrt{ST})$.

$\square$

## C   Technical Lemmas

### C.1   Lemma C.1

**Lemma C.1.** *If*

$$K\varepsilon/2 \le a\sqrt{K}\sqrt{\gamma(K,\delta)} + b\gamma(K,\delta)\log(K+1),$$

*where* $\gamma(K,\delta) = 2\log(SAHK/\delta) + (S-1)\log(e(1+K/(S-1)))$, *then*

$$K = \widetilde{\mathcal{O}}\left(\left(\frac{a^2}{\varepsilon^2} + \frac{b}{\varepsilon}\right)\log\left(\frac{1}{\delta}\right) + \frac{Sb}{\varepsilon}\right),$$

*where* $\widetilde{\mathcal{O}}(\cdot)$ *ignores all the term* $\log S, \log A, \log H, \log 1/\varepsilon$ *and* $\log\log(1/\delta)$.

*Proof.* If $K \le 4$, the lemma is trivial. Now assume $K \ge 4$, then $e(1+K) \le 4K$ and

$$\begin{aligned}
\gamma(K,\delta) &\le 2\log(SAHK/\delta) + S\log(e(1+K)) \\
&\le 2\log(SAHK/\delta) + S\log(4K) \\
&\le 2S\log(4K) + \log(SAH/\delta).
\end{aligned}$$

Then

$$\begin{aligned}
K\varepsilon/2 &\le a\sqrt{K}\sqrt{\gamma(K,\delta)} + b\gamma(K,\delta)\log(K+1) \\
&\le 2\max\{a\sqrt{K}\sqrt{\gamma(K,\delta)}, b\gamma(K,\delta)\log(K+1)\}.
\end{aligned}$$

**Case 1:**   If $K\varepsilon/2 \le 2a\sqrt{K}\sqrt{\gamma(K,\delta)}$, we can get

$$\begin{aligned}
K &\le \frac{16a^2}{\varepsilon^2}(2S\log(4K) + \log(SAH/\delta)) \\
&\le 2\max\left\{\frac{32Sa^2}{\varepsilon^2}\log(4K), \frac{16a^2}{\varepsilon^2}\log(SAH/\delta)\right\}.
\end{aligned}$$

**Subcase 1:**   If $K \le \frac{64Sa^2}{\varepsilon^2}\log(4K)$ by Lemma C.2, we can get $K = \widetilde{\mathcal{O}}(\frac{Sa^2}{\varepsilon^2})$.

**Subcase 2:**   If $K \le \frac{16a^2}{\varepsilon^2}\log(SAH/\delta)$, we complete the proof.

**Case 2:**   If $K\varepsilon/2 \le 2b\gamma(K,\delta)\log(K+1)$, we can get

$$\begin{aligned}
K &\le \frac{4b}{\varepsilon}\log(4K)(2S\log(4K) + \log(SAH/\delta)) \\
&\le 2\max\left\{\frac{4b}{\varepsilon}\log(4K)2S\log(4K), \frac{4b}{\varepsilon}\log(4K)\log(SAH/\delta)\right\}.
\end{aligned}$$

**Subcase 3:**   If $K \le \frac{8bS}{\varepsilon}\log^2(4K)$ By Lemma C.2, we can get $K = \widetilde{\mathcal{O}}(\frac{8bS}{\varepsilon})$.

**Subcase 4:** If $K \leq \frac{8b}{\varepsilon} \log(4K) \log(SAH/\delta)$, then by Lemma C.2, we can get $K = \widetilde{\mathcal{O}}(\frac{b}{\varepsilon} \log(1/\delta))$

From the four subcases, we complete the proof of Lemma C.1.

$\square$

## C.2 Lemma C.2

**Lemma C.2.** *If $K \leq Q \log^2(4K)$ with $Q \geq 4$, then we have $K = \widetilde{\mathcal{O}}(Q)$.*

*Proof.* Define $f(x) = \frac{x}{\log^2(4x)}$, then $f'(x) = \frac{\log^2(4x) - \log(4x)}{\log^4(4x)} > 0$ when $4x \geq 3$, and $f(x)$ is increasing over $x \geq 1$. Then we only need to prove there exists a constant $c_2 > 0$ such that

$$f(c_2 Q \log^2 Q) \geq Q.$$

In fact, if this inequality holds, any $K$ such that $K \leq Q \log^2(4K)$ should have $f(K) \leq Q \leq f(c_2 Q \log^2 Q)$, and then $K \leq c_2 Q \log^2 Q = \widetilde{\mathcal{O}}(Q)$. To prove this inequality, observe that

$$
\begin{aligned}
Q \log^2(4c_2 Q \log^2 Q) &= Q(\log 4c_2 Q + \log \log^2 Q)^2 \\
&\leq 2Q \log^2(4c_2 Q) + 2Q(\log \log^2 Q)^2 \\
&\leq 4Q \log^2(4c_2) + 4Q \log^2(Q) + 2Q \log^2 Q \\
&\leq (4 \log^2(4c_2) + 6) Q \log^2 Q.
\end{aligned}
$$

where the third inequality is because $\log^2 Q \leq Q$ for $Q \geq 4$. So if we choose $c_2$ to satisfy $c_2 \geq 4 \log^2(4c_2) + 6$, we have $Q \log^2(4c_2 Q \log^2 Q) \leq c_2 Q \log^2 Q$, which implies $f(c_2 Q \log^2 Q) \geq Q$. $\square$

# D Detailed Analysis of Assumption 4.1

In this section, we provide a more detailed analysis of the equivalence between $s_1 \notin \mathcal{U}_1$ and the existence of feasible policy. Indeed, if the agent gets into some state $s \in \mathcal{U}_h$ at step $h$, then any action can lead her to an unsafe next state $s' \in \mathcal{U}_{h+1}$. Recursively, any action sequence $a_h, a_{h+1}, \cdots, a_{H-1}$ can lead the agent to unsafe states $s_{h+1} \in \mathcal{U}_{h+1}, \cdots, s_H \in \mathcal{U}_H$. As a result, the agent cannot avoid getting into an unsafe state. Therefore, all feasible policies $\pi$ must satisfy that, at any step $h$, the probability for the agent in the state $s \in \mathcal{U}_h$ should be zero under policy $\pi$.

$$\mathbb{P}_h^\pi(s) = 0, \quad \forall s \in \mathcal{U}_h. \tag{21}$$

Recall that

$$A_h^{safe}(s) = \{a \in \mathcal{A} \mid \Delta_h(s,a) \cap \mathcal{U}_{h+1} = \emptyset\}.$$

From these definitions, a feasible policy $\pi$ should satisfy that $\pi_h(s) \in A_h^{safe}(s)$ for any safe state $s \notin \mathcal{U}_h$. Moreover, for any safe state $s_h \notin \mathcal{U}_h$, there exists at least one safe action $a_h \in A_h^{safe}(s_h)$, and all possible next states $s_{h+1} \in \Delta(s_h, a_h)$ satisfy $s_{h+1} \notin \mathcal{U}_{h+1}$. Recursively, there exists at least one feasible action sequence $a_h \in A_h^{safe}(s_h), \cdots, a_H \in A_H^{safe}(s_H)$ which satisfies that $s_{h'} \notin \mathcal{U}_{h'}$ for $h+1 \leq h' \leq H$. Hence the assumption for the existence of feasible policy can be simplified to $s_1 \notin \mathcal{U}_1$.

# E Experiment Setup

We perform each algorithm for 20000 episodes in a grid environment (Chow et al. (2018); Wei et al. (2022)) with 25 states and 4 actions (four directions), and report the average reward and violations across runs. In the experiment, we note that the algorithm Optpess-bouns (Efroni et al., 2020b) needs to solve a linear programming with size $O(SAH)$ for each episode, which is extremely slow. Thus we only compare all algorithms in a relatively small environment. In the Safe-RL-SW experiments, we choose $\delta = 0.005$ and binary 0-1 cost function. We set the safety threshold as 0.5 for both the CMDP algorithms and the SUCBVI algorithm.

For the Safe-RFE-SW experiments, we compare our algorithm SRF-UCRL with a state-of-the-art RFE algorithm RF-UCRL (Kaufmann et al., 2021) in an MDP with 11 states and 5 actions. We choose $\delta = 0.005$ and safety threshold of SRF-UCRL as 0.5. We run 500 episodes 100 times, and then report the average reward in each episode and cumulative step-wise violation. Also at each episode, we calculate the expected violation for the outputted policy and plot the curves. All experiments are conducted with AMD Ryzen 7 5800H with Radeon Graphics and a 16GB unified memory.