# OpenReview forum: "Provably Safe Reinforcement Learning with Step-wise Violation Constraints"
_NeurIPS.cc/2023/Conference — NeurIPS 2023 poster_

### Official Review · Reviewer_31rE · 2023-06-24

**Soundness:** 3 good
**Presentation:** 3 good
**Contribution:** 3 good
**Rating:** 7
**Confidence:** 4

**Summary:**

This paper studies safe RL with step-wise violation constraints, different from the popular CMDP with
an additive expectation cost constraint. The step-wise violation constraint is more suitable for safety-critical
systems. The authors propose an algorithm that provides violation and regret bound. They then further
develops an algorithm to learn a near-optimal safe policy and show its effectiveness in the experiments.



**Strengths:**

1) This paper studies an important problem. The paper is well-written and easy to follow.
The RL with step-wise violation constraint, as a formulation, is novel and more general than the popular CMDP with an additive expectation constraint.
2) The proposed approach looks correct and sound to me although I didn't check every proof.
3) The authors provide theoretical analyses of violation and regret bound.
4) The proposed safe RL algorithm achieves better performance than the existing baselines.

**Weaknesses:**

1. This paper may miss some reference for safe RL with step-wise violations such as

a) Wang, Y., Zhan, S. S., Jiao, R., Wang, Z., Jin, W., Yang, Z., ... & Zhu, Q. (2022).
Enforcing Hard Constraints with Soft Barriers: Safe Reinforcement Learning in Unknown Stochastic Environments.
arXiv preprint arXiv:2209.15090.

The authors have to at least discuss this paper in this work as the timepoint-level "hard chance constraint" looks very similar to the step-wise violations proposed by the authors.

2. In the MDP model, it is not clear to me whether the paper solves safe RL with continuous/discrete deterministic/stochastic
systems or environments. The transition set \Delta_h(s, a) could be an infinite set when considering continuous state space.

3. Essentially, my understanding of this paper is that they are dealing with discrete action and state space, and because of this,
the RL agent can infer which state and/or future state is unsafe to visit by building the A_safe set via dynamical programming.
which is the core of the proposed algorithms.


**Questions:**

I am using a Windows laptop and I cannot open the full_paper file (format) in the supplementary folder, so I cannot see the limitations of the approach.

**Limitations:**

I would like to authors to clarify the limitations of their formulation and approach in the response.

---

> ### Author Rebuttal · Authors · 2023-08-09
>
> Thanks for your time and effort in reviewing our paper! We really appreciate your positive comments about our work. Please find our responses to your comments below. We will be happy to answer any further questions you may have.
>
>
> **1. Discuss with [3].**
>
> [3] considers the same step-wise constraint as ours. They solve step-wise constraints via  learning a generative-model-based soft barrier function. To be more specific, they use a soft barrier function to encode the constraint of the surrogate model provided by the generative model. They learn the generative model as a discrete-time SDE.
>
>    There are two major differences between [3] and our work: (a) They can only solve the continuous dynamics (stated in Assumption 3.1 in [3]), while we consider discontinuous tabular MDPs. (b) They focus on the empirical side and do not obtain any theoretical guarantees, while we focus on the theoretical side and provide rigorous regret analysis and lower bounds.
>
>
> **2. Continuous space**
>
>
>
>    We agree that it is interesting and challenging to extend our framework to continuous state space since $\Delta(s,a)$ is infinite there. [2] avoids this problem by assuming $\Delta(s,a)$ is known. An important future direction is to study whether partial information of $\Delta(s,a)$ is sufficient, instead of the entire $\Delta(s,a)$. Moreover, we want to emphasize that previous works [1,2] cannot solve the safety problem on tabular MDPs: They assume the linear feature set is star and convex (Assumption 5 in [1] and Assumption 2 in [2]), which does not hold in tabular MDPs since the feature set in tabular MDP consists of one-hot vectors.
>
>    We emphasize that our problem is challenging and novel. In fact,  [3] states that discontinuous  MDP  still remains challenging, even if the setting of continuous state space is solved. Thus, our results for tabular and discontinuous MDPs are significant.
>
>
>
> **3. Cannot open the full paper file**
>
>    The "full$\\_$paper" file can be opened by adding a ".pdf" suffix. Thanks for pointing out this issue. We will fix it in our revision.
>
>    **Here we describe the limitation of our paper.**
>    First, Algorithm 1 will suffer a non-zero step-wise violation in the beginning. This non-zero violation is unavoidable, because the agent has no information about unsafe states to begin with. Second, in some real-world environments, the number of states and actions can be large and even infinite. Thus, the complexity of our algorithm can be high. To overcome this challenge, one interesting direction is to extend our work to the function approximation setting, such as linear MDP [4].
>
>
> **References:**
>
> [1]. Amani S, Thrampoulidis C, Yang L. Safe reinforcement learning with linear function approximation, ICML, 2021.
>
>
> [2]. Shi M, Liang Y, Shroff N. A Near-Optimal Algorithm for Safe Reinforcement Learning Under Instantaneous Hard Constraints, arXiv preprint, 2023.
>
> [3]. Wang Y, Zhan S S, Jiao R, et al. Enforcing hard constraints with soft barriers: Safe reinforcement learning in unknown stochastic environments, ICML 2023.
>
> [4]. Jin C, Yang Z, Wang Z, et al. Provably efficient reinforcement learning with linear function approximation, COLT 2020.

---

> > ### Comment · Reviewer_31rE · 2023-08-12
> >
> > Thanks for the rebuttals. In your future revision, please make sure to clarify that this paper considers a tabular MDP in the problem formulation.

---

> > > ### Comment · Reviewer_31rE · 2023-08-14
> > >
> > > I am happy to increase my score to a 7.

---

> > > > ### Author Response · Authors · 2023-08-16
> > > >
> > > > Thank you very much for increasing the score! We will clarify the tabular MDP setting in our revision.

---

### Official Review · Reviewer_ZGMh · 2023-07-03

**Soundness:** 2 fair
**Presentation:** 2 fair
**Contribution:** 3 good
**Rating:** 5
**Confidence:** 3

**Summary:**

This paper formulates and studies a strict step-wise violation constraint reinforcement learning problem, where a non-negative state-dependent cost is cumulated at each step. They show lower bounds for regret and safety violations. A model-based algorithm that matches the lower bound is provided, while another O(1) gap-dependent violation is provided too. They also study the reward-free problem and provide an algorithm that has a sample complexity with a leading term matching the SOTA of a no-constraint case.

**Strengths:**

The step-wise violation constraint problem is meaningful. This paper is technically sound. The proofs are correct as far as I can tell. The empirical result looks good.

**Weaknesses:**

1. The idea of the unsafe set can only apply to state-dependent safety problems. I doubt this technique can apply to a (state, action)-dependent safety problem. Additionally, it is weird that the reward is (state, action)-dependent, but the cost is state-dependent only.
2. It is weird that the reward-free can give cost feedback.
3. Definition 6.1 is confusing. I guess from your proof that \pi in (4) means all feasible policies defined by \Delta and U?
4. line 545, '<' should be '>'?



**Questions:**

See weakness.

**Limitations:**

Part of the limitations are shown. The authors don't discuss the limitations of state-dependent cost as well as the weird reward-free setting.

---

> ### Author Rebuttal · Authors · 2023-08-09
>
> Thanks for your time and effort in reviewing our paper!  Please find our responses to your comments below. We will be happy to answer any further questions you may have.
>
>
> **1. Can the cost be (state, action)-dependent?**
>
> The reason why we choose the state-dependent cost is to follow the setting in previous works [1,2,3] for step-wise constraints.
>
>    Extending our framework to (state, action)-dependent cost is not difficult. Intuitively, we can define a state $s$ is unsafe when $\max_{a \in \mathcal{A}}c(s,a) \le \tau$. Then, in each state $s$, we can also use the uniform exploration to choose the action and estimate $c(s,a)$. It only leads to an extra $A$ factor in regret and violation.
>
>
> **2. It is weird that the reward-free can give cost feedback.**
>
>    The motivation of RFE is that, since the reward functions are often engineered and changed in practice, we need an algorithm to explore the environment and work well for different rewards [4]. In many real-world scenarios such as robot control, the cost function depends on the environment, which does not change frequently. Hence, it is natural that we can receive the cost feedback without the reward signal. This setting is also considered in previous works for safe RFE, e.g., [5].
>
> **3. Definition 6.1 is confusing**
>
> Thanks for pointing out this confusion. $\pi$ in Eq.(4) denotes any feasible policies in $\Pi$ with respect to true $\Delta(s,a)$ and $\mathcal{U}$. Note that $\Pi^K$ is feasible with respect to $\hat{\Delta}^K(s,a)$ and $\mathcal{U}^k_H$, thus $\Pi^K\subseteq \Pi$ by  optimism. We will improve the explanation in our revision.
>
> **4. Typo**
>
>    The equation under Line 545 is $\ge$. We will fix this typo in our revision.
>
> **References:**
>
> [1]. Wachi, A., Sui, Y., Yue, Y., and Ono, M. Safe exploration and optimization of constrained
> MDPs using Gaussian processes. AAAI 2018.
>
> [2]. Wachi A, Wei Y, Sui Y. Safe policy optimization with local generalized linear function approximations[J]. Advances in Neural Information Processing Systems, 2021
>
> [3]. Wang Y, Zhan S S, Jiao R, et al. Enforcing hard constraints with soft barriers: Safe reinforcement learning in unknown stochastic environments, ICML 2023.
>
> [4]. Jin C, Krishnamurthy A, Simchowitz M, et al. Reward-free exploration for reinforcement learning, ICML 2020.
>
> [5]. Huang R, Yang J, Liang Y. Safe exploration incurs nearly no additional sample complexity for reward-free rl, 2022.

---

> > ### Comment · Reviewer_ZGMh · 2023-08-16
> >
> > Thank you for the rebuttals. I still have some concerns after reading the rebuttal and I will keep my rating.
> >
> > 1. If the framework is extended to (state, action)-dependent cost as defined by authors, then it seems no hope to handle the large action case even if function approximation is used since the definition of unsafe states always requires maximize over all actions.
> >
> > 2. I am still not fully convinced by the reasons that the cost feedback can be given in the reward-free setting. In some cases, the reward function can also depends on the environment, which does not change frequently.

---

> > > ### Author Response · Authors · 2023-08-16
> > > **Response to Reviewer ZGMh**
> > >
> > > Thank you for the response!
> > >
> > > **1.If the framework is extended to (state, action)-dependent cost as defined by authors, then it seems no hope to handle the large action case even if function approximation is used since the definition of unsafe states always requires to maximize over all actions.**
> > >
> > > If we extend to (state, action)-dependent cost, it indeed needs to explore all actions to receive the safety feedback of each action. One possible avenue to avoid this exploration is to posit that the cost function exhibits a certain favorable structure, (e.g. $c(s,a)= \theta^T \varphi(s,a)$ for some $\theta \in \mathbb{R}^d$ and known features $\varphi(s,a) \in \mathbb{R}^d$, as in existing works [1,8].) and then estimate the cost for all $(s,a)$ exploiting the structure without exploring all actions. This is an interesting future direction.
> > >
> > >
> > >
> > > **2. I am still not fully convinced by the reasons that the cost feedback can be given in the reward-free setting. In some cases, the reward function can also depend on the environment, which does not change frequently.**
> > >
> > > The RFE setting is motivated by the fact that in practice, the reward function can be difficult to specify and change frequently by manual crafting [2,3], or there can be multiple rewards of interest in the same environment [4].
> > >
> > > In contrast, the cost function can heavily depend on the environment such as barrier placements [5] and the gesture of the robot [6]. Hence, it can be immutable and knowable. Previous safe RFE work [7] also receives the cost feedback during the exploration phase. Moreover, from a theoretical perspective, access to information about the cost function becomes imperative. Devoid of this information, ensuring safety constraints during the exploration phase becomes an impractical pursuit.
> > >
> > >
> > >
> > > We thank the reviewer again for the detailed and valuable comments. Please let us know if you have any further questions. We will be happy to answer them. If you find our response satisfying, we wonder if you could kindly consider raising the score rating of our work?
> > >
> > > Thank you very much!
> > >
> > > **References:**
> > >
> > > [1]. Amani S, Thrampoulidis C, Yang L. Safe reinforcement learning with linear function approximation, ICML, 2021.
> > >
> > >
> > >
> > > [2]. Leike J, Krueger D, Everitt T, et al. Scalable agent alignment via reward modeling: a research direction, preprint 2018.
> > >
> > > [3]. Fu J, Singh A, Ghosh D, et al. Variational inverse control with events: A general framework for data-driven reward definition, NIPS 2018.
> > >
> > > [4]. Wu J, Braverman V, Yang L. Accommodating picky customers: Regret bound and exploration complexity for multi-objective reinforcement learning, NIPS 2021.
> > >
> > > [5]. Thananjeyan B, Balakrishna A, Nair S, et al. Recovery rl: Safe reinforcement learning with learned recovery zones, IEEE Robotics and Automation Letters, 2021.
> > >
> > > [6]. Thomas G, Luo Y, Ma T. Safe reinforcement learning by imagining the near future, NIPS 2021.
> > >
> > > [7]. Huang R, Yang J, Liang Y. Safe exploration incurs nearly no additional sample complexity for reward-free rl, ICLR 2023.
> > >
> > > [8]. Wachi et al. Safe policy optimization with local generalized linear function approximations. 2021.

---

> > > > ### Comment · Reviewer_ZGMh · 2023-08-16
> > > >
> > > > Thank you for your reply. Even with the linear structure, it seems that the definition of safe set stills requires to maximize over the entire action space. I understand the requirement of cost information theoretically. However, in practice, this setting doesn't make sense in some cases. For example, there are multiple costs of interest in the wireless network, e.g., bandwidth and expense.

---

> > > > > ### Author Response · Authors · 2023-08-17
> > > > >
> > > > > (a). We agree that we need to compute the maximum value $\max_{a \in A} c(s,a)$. In fact, in the realm of linear MDP, in terms of computation, all previous works need to calculate the maximization over the action space,  e.g., [1,2,3]. However, there is no need to actually explore (i.e., actually take) all the actions.
> > > > >
> > > > > (b). In the wireless network example, even if there are multiple costs, e.g., bandwidth and expense, these costs and constraints can be known beforehand. Subsequently, we can treat this as a multi-constraint problem, and use a vector-valued cost function to simultaneously represent multiple cost functions [6].
> > > > >
> > > > > Moreover, reward serves as a way for us to specify the objectives under different tasks, and can be changed frequently by manual crafting [4,5]. Thus, it becomes crucial to devise an algorithm that effectively learns the underlying model and attains commendable performance across diverse reward scenarios. On the other hand, cost often depends on the environment and is an inherent attribute of the environment, rather than being manually designated. Hence, it is reasonable to receive the cost feedback during the exploration phase.
> > > > >
> > > > >
> > > > >
> > > > >
> > > > >
> > > > >
> > > > >
> > > > > [1]. Jin C, Yang Z, Wang Z, et al. Provably efficient reinforcement learning with linear function approximation, COLT 2020.
> > > > >
> > > > > [2]. Hu P, Chen Y, Huang L. Nearly minimax optimal reinforcement learning with linear function approximation, ICML 2022.
> > > > >
> > > > > [3]. Amani S, Thrampoulidis C, Yang L. Safe reinforcement learning with linear function approximation, ICML, 2021.
> > > > >
> > > > > [4]. Leike J, Krueger D, Everitt T, et al. Scalable agent alignment via reward modeling: a research direction, preprint 2018.
> > > > >
> > > > > [5]. Fu J, Singh A, Ghosh D, et al. Variational inverse control with events: A general framework for data-driven reward definition, NIPS 2018.
> > > > >
> > > > > [6]. Yu T, Tian Y, Zhang J, et al. Provably efficient algorithms for multi-objective competitive rl, ICML 2021.

---

### Official Review · Reviewer_ryzU · 2023-07-04

**Soundness:** 3 good
**Presentation:** 3 good
**Contribution:** 3 good
**Rating:** 6
**Confidence:** 3

**Summary:**

The authors propose a new formulation for the safe RL problem Safe-RL-SW, whose violation constraints are step-wise, different from the existing work. A model-based general algorithmic framework SUCBVI is proposed and the theoretical guarantees on the upper bound and lower bound of its regret are provided. The authors also propose a similar reward-free safe RL problem with step-wise violation constraints (Safe-RFE-SW) along with an algorithm SRF-UCRL, the theoretical upper bound on whose regret is provided. It is claimed to be the first result on step-wise violation constraints in the RFE setting. Experiments are provided in support of the theoretical results.

**Strengths:**

The authors propose new problem formulations for safe RL as well as algorithms under these formulations. Theoretical results are provided for the proposed algorithms. Overall, the new problem formulations seem original and the results seem significant.

**Weaknesses:**

How the results in this work compare with the existing work is somewhat unclear, despite some discussion in the Related Work Section. In Section 2, the authors mention two closely-related work Amani et al. (2021) and Shi et al. (2023) that also consider step-wise violation and state that more detailed comparisons can be found in Appendix B, but I did not find any comparison with these two works in Appendix B. I strongly suggest that the authors compare with all related works and clearly state how this work differentiates from them in the revision. For example, do all CMDP works define violation with $C'(K)$ in Line 136? Can you compare with the CMDP works that do not use expected violation like $C'(K)$, if there is any? As for Amani et al. (2021) and Shi et al. (2023), please explain whether there is any difference in your formulations and how your results compare with theirs, if your setting is comparable with theirs.

**Questions:**

I hope the authors can provide clarification for a few questions:

1. What is the role of $n$ in Theorem 5.2? Why considers $M_1, \cdots, M_n$ in this theorem instead of just focusing on the one MDP that establishes the lower bound?

2. Do all CMDP works define violation with $C'(K)$ in Line 136? Is there any CMDP work that do not use $C'(K)$ and have a violation definition that is more comparable to the step-wise violation in this work?

3. In the plot of "Step-wise Violation in Safe-RL-SW" of Figure 1, the cumulative step-wise violation curve for SUCBVI seems to grow much more slowly than $\sqrt{T}$. However, shouldn't we expect something $\Omega(\sqrt{ST})$, especially given your lower bounds in Section 5?

**Limitations:**

The limitation is currently discussed in Appendix A. I encourage the authors to move part of it in the conclusion so that it is more conspicuous.

---

> ### Author Rebuttal · Authors · 2023-08-09
>
> Thanks for your time and effort in reviewing our paper.  We will be happy to answer any further questions you may have.
>
> **1. Comparison with existing works**
>
>  We provide a brief summary of existing papers with instantaneous constraints and list the assumptions.
>
> **(1). Gaussian Process (GP) Structure**
>
> [1,2] propose a GP-based algorithm, which assumes that (a) the transition is deterministic and known, and (b) the reward and cost functions are modeled by GPs. By using this particular structure, they can infer the safety cost by estimating the parameters.
>
> **(2). Unbounded exploring time**
>
> [3] assumes the reward and cost functions have a generalized linear structure. Their algorithm explores in a safe space until a time $t^*$ when the agent explores sufficiently. However, the upper bound of the exploring time $t^*$ is not given in their paper. Under the tabular MDP setting, $t^*$ can be infinite since the linear feature vectors consist of one-hot vectors.
>
> **(3). Safe action, convex feature set and known transition set**
>
> [4] considers the reward and cost functions to have a linear structure. It makes two assumptions: (a) There exists a safe action in each state, which avoids the agent go to a potentially unsafe state. (b) The feature set is a star convex set, which helps them change actions continuously. However, this makes their works infeasible in tabular MDPs: The feature set in tabular MDPs consists of one-hot vectors and is not a star convex set. Hence, the work [4] cannot solve our problem. [6] considers safe RL in linear mixture MDPs. It also contains assumption (b), making it infeasible in tabular MDPs. Moreover, although they do not have assumption (a), it assumes the transition set $\Delta(s,a)$ is known, which is not needed in our paper. Thus our paper is more challenging since we need to estimate $\Delta(s,a)$ in our algorithm adaptively.
>
> **(4). Safety feedback oracle**
>
> [5] considers safe RL problems with binary safety feedback. They also assume there is a safe action in each state. Moreover, they can receive safety feedback of $(s,a)$ by calling an oracle without taking action $a$ at state $s$.
>
> There are also other related works that do not provide regret or sample complexity analysis [7,9], so we do not provide a technical comparison. In contrast, our paper does not require any of these assumptions.
>
>
> **2. Other CMDP formulation**
>
> Most CMDP works focus on the **expected violation**, i.e., $C'(K) = \sum_{k=1}^K (E[\sum_{h=1}^H c(s_h^k,a_h^k)]-\mu)$
> to define their constraints. [10,11] consider the **clipped expected violation**
> $C''(K)=\sum_{k=1}^K (E[\sum_{h=1}^H c(s_h^k,a_h^k)]-\mu)_+$, which is a slightly stricter constraint than the expected violation.
>
> Different from previous works, we consider the step-wise violation $$C(K)=\sum_{k=1}^K \sum_{h=1}^H (c(s_h^k)-\tau)_+,$$
>
> which enforces the safety at each step. The clipped expected violation $C''(K)$ in [10,11] cannot guarantee the safety at each step with certainty, since in the definition of $C''(K)$, the average violation at each step $c(s_h^k,a_h^k)-\mu/H$ can be positive or negative, and they can cancel out to get $E[\sum_{h=1}^H c(s_h^k,a_h^k)]\le \mu$.
>
> **3. Comparison for [4,6].**
>
> As we mentioned in Reply 1, [4,6] cannot work for tabular MDPs. Moreover, [4] requires an assumption that there exists a safe action in each state, while [6] needs known $\Delta(s,a)$. In our work, we do not require these assumptions. Hence, their works and ours cannot be directly compared.
>
> **4. Why consider n MDPs instead of focusing on one MDP for the lower bound?**
>
> If we just focus on one MDP M, it is impossible to derive a lower bound for M: If the algorithm guesses the optimal policy $\pi^*$ for a MDP correctly and takes $\pi^*$ all the time, it can achieve zero regret and violation on M. Therefore, to construct the lower bound, we must consider multiple MDPs, so that no algorithm can always take the optimal policies for all MDPs because the optimal policies are different.  Then, one can prove that the algorithm does not work well in one of them. This construction idea has been used in many prior works for different problems, e.g., [8,12].
>
> **5. The cumulative violation curve grows more slowly than $\sqrt{T}$.**
>
> Theorem 4.2 shows that if the safety gap $C_\text{gap}>0$, we can get a bounded $\tilde{O}(S/C_\text{gap}+ S^2AH^2)$ violation. In our experiment, the safety gap is large, and our curve matches this result.
> However,  the lower bound Theorem 5.1 is constructed by the worst instance with safety gap $C_{\text{gap}}= O(\sqrt{S/T})$ in the construction. This safety gap is small and changes for different numbers of episodes $T$, which is not the setting of our experiment.
>
> **References:**
>
> [1]. Turchetta et al. Safe exploration in finite Markov decision processes with gaussian processes. 2016.
>
> [2]. Wachi et al. Safe exploration and optimization of constrained MDPs using Gaussian processes. 2018.
>
> [3]. Wachi et al. Safe policy optimization with local generalized linear function approximations. 2021.
>
> [4]. Amani et al. Safe reinforcement learning with linear function approximation, 2021.
>
> [5].Bennett et al. Provable Safe Reinforcement Learning with Binary Feedback, 2023.
>
> [6]. Shi et al. A Near-Optimal Algorithm for Safe Reinforcement Learning Under Instantaneous Hard Constraints, 2023.
>
> [7]. Wang et al. Enforcing hard constraints with soft barriers: Safe reinforcement learning in unknown stochastic environments, 2023.
>
> [8]. Simchowitz et al. Non-asymptotic gap-dependent regret bounds for tabular mdps, 2019.
>
> [9]. Thomas et al. Safe reinforcement learning by imagining the near future, 2021.
>
> [10]. Efroni et al. Exploration and exploitation in CMDP, 2020.
>
> [11]. Simao et al. AlwaysSafe: Reinforcement learning without safety constraint violations during training, 2021.
>
> [12] Domingues et al. Episodic reinforcement learning in finite mdps: Minimax lower bounds revisited, ALT 2021.

---

> > ### Comment · Reviewer_ryzU · 2023-08-13
> >
> > I appreciate the rebuttal from the authors. When you revise the paper, please make sure to discuss the comparison with other works and any implications of the proposed problem formulation sufficiently in the main text. I have raised my score.

---

> > > ### Author Response · Authors · 2023-08-16
> > >
> > > Thank you very much for raising the score! We will discuss the comparison with other works and the implications of the formulation sufficiently in our revision.

---

### Official Review · Reviewer_83sH · 2023-07-21

**Soundness:** 4 excellent
**Presentation:** 4 excellent
**Contribution:** 3 good
**Rating:** 5
**Confidence:** 4

**Summary:**

This paper considers online RL problem for an MDP with stage-wise constraints. The stage-wise constraints basically specify the set of unsafe states which must be avoided at all times. While there is a lot of recent work on CMDPs, this formulation, which is actually more relevant is less studied. The authors propose a variant of UCBVI algorithm, and show that it achieves O(\sqrt{SATH^3}) regret which is order optimal except in H. It also achieves sublinear constraint violation, which is independent of T when it is gap-dependent. They also present a reward-free exploration algorithms with sublinear regret and constraint violation.



**Strengths:**

This seems one of the few results on Online RL for MDP with stage-wise constraints. Attempt at numerical evaluation is commendable.

**Weaknesses:**

Unfortunately, the results are weaker than what have been achieved recently for CMDP problems: some of the formulations only consider clipped violation functions, and still achieve bounded constraint violation. The results for reward-free learning are actually much weaker as it only achieves sublinear constraint violation. Furthermore, novelty in the algorithm is limited based as it is on UCBVI, a well-trodden path. Recent work on online learning for CMDPs has several interesting algorithmic ideas which the authors could potentially use to propose algorithms for sharper results.

**Questions:**

Could you use some of the recent work of Lei Ying and Dileep Kalathil to design algorithms that achieve even sharper results. I believe it is possible without too much extra work.

**Limitations:**

N/A.

---

> ### Author Rebuttal · Authors · 2023-08-09
>
> Thanks for your time and effort in reviewing our paper!  Please find our responses to your comments  below. We will be happy to answer any further questions you may have.
>
> **1. Unfortunately, the results are weaker than what has been achieved recently for CMDP problems: some of the formulations only consider clipped violation functions and still achieve bounded constraint violation.**
>
> To the best of our knowledge, [1,5] are the most related papers that consider the clipped violation $C''(K)=\sum_{k=1}^K (\mathbb{E}[\sum_{h=1}^H c(s_h^k,a_h^k)]-\mu)_+$ for a safety threshold $\mu$. However, our results are not weaker than theirs. In particular, our step-wise constraint induces a stricter requirement as it requires the agent to stay safe **at each step with certainty** rather than in expectation over each episode.
>
> We will be happy to explain the differences if the reviewer has concerns about any other references.
>
> **2. The results for reward-free learning are actually much weaker as it only achieves sublinear constraint violation.**
>
> First of all, note that similar to the regret minimization setting, we can derive a constant constraint violation if the safety gap $\mathcal{C}_\mathrm{gap}>0$.
>
> To our best knowledge, there is only one work [4] considering safety problems in RFE. They consider RFE in the CMDP setting with expected constraints, and achieve zero violation during exploration. Compared to them, we consider a stricter step-wise constraint that requires the agent to stay safe **at each step with certainty**, and provide bounded violation guarantees.
>
> **3. Novelty in the algorithm is limited based as it is on UCBVI, a well-trodden path.**
>
> Although our algorithm is motivated by UCBVI, our algorithm design and analysis are novel. Compared to unconstrained RL,  we use a novel dynamic program to estimate the set of potentially unsafe states and plan for the future in each episode, avoiding going to an unsafe state in the future inevitably. In addition, since the potentially unsafe states are changing in each episode, we need to further consider the uncertainty brought by the inaccurately estimated unsafe states in regret analysis.
>
> Moreover, we consider safety problems in the RFE setting, which is an important and challenging problem in RL. Compared to unconstrained RFE problems, the agent needs to both stay safe during the exploration phase and output a nearly safe policy afterward.  Previous work for RFE in CMDP [4] cannot be directly applied to solve this problem, since they only consider expected violations during the exploration phase, and thus cannot guarantee safety for each step with certainty. We tackle this challenge by proposing a new uncertainty function defined in Eqs. (8) and (9) to upper bound both the violation during exploration and the expected violation for the output policy.
>
> **4. Recent work of Lei Ying and Dileep Kalathil**
>
> We find two related works by these two authors. [2] designs model-free algorithms for non-stationary CMDPs, where the main framework builds upon the Triple-Q algorithm.  [3] considers doubly optimistic and pessimistic exploration to achieve zero violation when there is a safe baseline policy. These two papers are interesting but they both focus on the expected violation $C'(K)=\sum_{k=1}^K (\mathbb{E}[\sum_{h=1}^H c(s_h^k,a_h^k)]-\mu)$. Thus, their results cannot solve our step-wise constraint problems. In addition, we extend our algorithmic framework to solve the safety problem in RFE for the exploration phase and the final output policy, which is not considered in their papers.
>
> [1]. Efroni Y, Mannor S, Pirotta M. Exploration-exploitation in constrained mdps. preprint, 2020.
>
> [2]. Wei H, Ghosh A, Shroff N, et al. Provably Efficient Model-Free Algorithms for Non-stationary CMDPs, AISTATS 2023.
>
> [3]. Bura A, HasanzadeZonuzy A, Kalathil D, et al. DOPE: Doubly optimistic and pessimistic exploration for safe reinforcement learning, NIPS 2022.
>
> [4]. Huang R, Yang J, Liang Y. Safe exploration incurs nearly no additional sample complexity for reward-free rl, preprint 2022.
>
> [5]. Simao et al. AlwaysSafe: Reinforcement learning without safety constraint violations during training,2021

---

> > ### Comment · Reviewer_83sH · 2023-08-21
> >
> > I know the difference between stage-wise and time-averaged constraints. I understand [3], etc. handle constraint violation in expectation. Since your requirement it stage-wise, it effectively is tantamount to constraining the policy space at each stage so that the constraint is satisfied. I think it should be possible to guarantee, zero or bounded constraint violation in your space as well. I will keep my score.

---

### Official Review · Reviewer_RuZi · 2023-07-26

**Soundness:** 2 fair
**Presentation:** 3 good
**Contribution:** 3 good
**Rating:** 6
**Confidence:** 4

**Summary:**

The paper studies an episodic constrained reinforcement learning problem with step-wise constraints on states. The authors first extend the classical UCB-VI to step-wise constraints and prove the sub-linear optimality gap and step-wise constraint violation. A lower bound is also provided to show optimal dependence on the state space size and the number of episodes. Second, the authors present a reward-free algorithm that takes nearly optimal sample complexity and subliner step-wise constraint violation during exploration. Some numerical experiments are provided to show the effectiveness of the proposed algorithms.

**Strengths:**

**originality**

- The problem formulation follows the existing tabular constrained MDP with instantaneous constraints. The safety constraint used in this work is an inequality of a function of state. The learning objective is to minimize regret and step-wise violation. The instantaneous constraint is also studied in other papers, e.g., Safe Policy Optimization with Local Generalized Linear Function Approximations and Provable Safe Reinforcement Learning with Binary Feedback.

-  The only problem assumption is the existence of an initial state that is not in the unsafe set. This assumption seems to be the weakest in the literature. Since the feasible policy has to satisfy the constraint at every step, transition dynamics can't guarantee the feasibility of the next state no matter the actions taken. So, it might not be intuitive that the feasibility has nothing to do with transition dynamics.

- In the first algorithm, the authors extend the existing UCBVI by incorporating the estimation of safe and unsafe sets, and prove sublinear regret and step-wise violation. It is useful if the authors could discuss how this work differs from previous applications of UCBVI to constrained MDPs, e.g., (Amani et al., 2021), which studies linear function approximation and offers zero violation.

- In the second algorithm, the authors generalize the existing reward-free algorithm to step-wise constraints, and prove nearly-optimal sample complexity.

**quality**

- Most statements are clarified by remarks or proofs.

- It is useful if the authors can make a more technical comparison with existing works since this paper is more theoretical.

- In experiments, it might be not very fair to compare other methods that study value-based constraints.

- It is less discussed about the weaknesses of the proposed method. It is useful to discuss if the proposed methods handle function approximation and provide zero constraint violation.


**clarity**

- The main results are delivered clearly, and the paper is organized well.

**significance**

- It is important to develop RL algorithms that can learn step-wise constraints. The authors have extended two known unconstrained RL algorithms to tabular constrained MDPs with step-wise constraints. However, step-wise constraints have been tackled in several other works, which warrant detailed comparison to show the significance of this paper.

- The provided theoretical analysis and guarantees are more expected. It is less discussed new challenges of extending the existing unconstrained analysis.

- The provided experiments do not seem to reflect the performance of the proposed algorithms. There is no variance in learning curves. Since all algorithms directly converge, exploration does seem to be an issue for convergence, which does not explains the adaptivity of online RL algorithms.

**Weaknesses:**

- It is useful to make a more detailed comparison with existing works on constrained MDP with instantaneous constraints, especially problem assumptions. Discussing some related applications is also useful.

- The technical contributions can be better explained by comparing the constrained and unconstrained ones. It is useful to highlight the main technical challenges.

- It is useful to discuss extending to large problems using function approximation and providing better violation guarantees.

- For experiments, it is important to make a fair comparison with existing works on constrained MDPs with instantaneous constraints. It is also important to experiment algorithmms in a correct way by showing convergence and exploration randomness.

**Questions:**

Some questions are stated in Strengths and Questions.

Here are some other questions.

- How the confidence bonus in Algorithm 1 is constructed?

- Is there a lower bound for Algorithm 2 that shows the hardness of step-wise constraints?

- To see how the proposed algorithms deal with step-wise constraints, can the authors plot the instantaneous constraint violation?

**Limitations:**

Yes.

---

> ### Author Rebuttal · Authors · 2023-08-09
>
> Thanks for your time and effort in reviewing our paper!  We will be happy to answer any further questions you may have.
>
> **1. Comparison with existing works and applications**
>
> We provide a brief summary of existing papers with instantaneous constraints and list their assumptions.
>
> **(1). Gaussian Process (GP) Structure**
>
> [1,2] propose a GP-based algorithm, which assumes that (a) the transition is known, and (b) the reward and cost functions are modeled by GPs. Based on the GP structure, they can infer the safety cost by estimating the parameters.
>
> **(2). Unbounded exploration time**
>
> [3] assumes the reward and cost have a generalized linear structure. Their algorithm explores in a safe space until a time $t^*$ when the agent explores sufficiently. However, their is no upper bound of the exploration time $t^*$ in [3]. For tabular MDP, $t^*$ can be infinite since the linear feature vectors consist of one-hot vectors.
>
> **(3). Safe action, convex feature set and known transition set**
>
> [4] considers the reward and cost functions to have a linear structure. It makes two assumptions: (a) There exists a safe action in each state, which avoids the agent go to a potentially unsafe state. (b) The feature set is a star convex set, which helps them change actions continuously. However, this makes their works infeasible in tabular MDPs: The feature set in tabular MDPs consists of one-hot vectors and is not a star convex set. Hence, the work [4] cannot solve our problem. [6] considers safe RL in linear mixture MDPs. It also contains the assumption (b), making it infeasible in tabular MDPs. Moreover, although it does not require assumption (a), it assumes the transition set $\Delta(s,a)$ is known, which is not needed in our paper. Thus our problem is more challenging since we need to estimate $\Delta(s,a)$ adaptively.
>
> **(4). Safety feedback oracle**
>
> [5] considers safe RL with binary safety feedback. They also assume that there is a safe action in each state. Moreover, they can receive safety feedback of $(s,a)$ by calling an oracle without taking action $a$ in state $s$.
>
> There are other works that do not provide regret analysis [7,8]. Compared to the above works, our paper does not require any of their assumptions.
>
> **Applications**
>
> For robotic control in complex environments, it is crucial to prevent the robot from hitting walls or falling into pools at all time. Our algorithm can be applied to train the robot and ensure that it stays safe at each step.
>
> **2. Technical comparison with existing works and challenges/novelty**
>
> (i) All previous works require strong assumptions (a) to help the agent infer the safety cost without going to an unsafe state or taking an unsafe action, and (b) to prevent the agent from getting stuck in a state. We do not require these assumptions. Instead, we estimate the cost and transition set $\Delta(s,a)$, and update the estimated set of potentially unsafe states by a novel dynamic program adaptively. We also consider the uncertainty of inaccurate unsafe states in regret analysis.
>
> (ii) We consider the safe RFE setting. Compared to unconstrained RFE, the safety constraints make it harder for the agent to explore the environment.  We propose a new uncertainty function to bound the step-wise violation during exploration and the expected violation for the output policy  (Line 261).
>
> (iii) We establish lower bounds to demonstrate that Theorem 4.2 is minimax-optimal.  To the best of our knowledge, this is the first lower bound that shows the tradeoff between regret and violation for safe RL.
>
> **3. Function approximation and zero constraint violation**
>
> For the function approximation, the large state space makes the estimation of $\Delta(s,a)$ infeasible. [6] overcomes this difficulty by assuming $\Delta(s,a)$ is known. One direction is to study whether partial information about $\Delta(s,a)$ is sufficient. For the zero constraint violation, it is infeasible without additional assumptions since we can only receive the safety feedback by going to that state. Hence it is unavoidable to get in an unsafe state.
>
> **4. Experiments for instantaneous constraints and variance curves**
>
> Since all works for instantaneous constraints either require strong assumptions or are infeasible in discontinuous tasks, these works and ours cannot be directly compared.
>
> As for the variance curve, see Figure 1 in the global response. In Figure 1, We plot the first 2000 episodes since the variance is evident and most of the algorithms converge (except Triple-Q, which requires a large $T$ in theoretical results).
>
> **5. The confidence bonus in Algorithm 1**
>
> We provide the definitions of confidence bonuses in Theorem 4.2 (Line 202).
>
> **6. Lower bound for Algorithm 2**
>
> A lower bound for Algorithm 2 (RFE) remains open. This is a very interesting direction, and we plan to study it in future work.
>
> **7. Plot the instantaneous constraint violation**
>
> We provide the curve of instantaneous violation in the second figure of Figure 1 in our paper.
>
> References:
>
> [1]. Turchetta et al. Safe exploration in finite Markov decision processes with gaussian processes. 2016
>
> [2]. Wachi et al. Safe exploration and optimization of constrained MDPs using Gaussian processes. 2018.
>
> [3]. Wachi et al. Safe policy optimization with local generalized linear function approximations. 2021.
>
> [4]. Amani et al. Safe reinforcement learning with linear function approximation, 2021.
>
> [5].Bennett et al. Provable Safe Reinforcement Learning with Binary Feedback, 2023.
>
> [6]. Shi et al. A Near-Optimal Algorithm for Safe Reinforcement Learning Under Instantaneous Hard Constraints, 2023.
>
> [7]. Wang et al. Enforcing hard constraints with soft barriers: Safe reinforcement learning in unknown stochastic environments, 2023.
>
> [8]. Thomas et al. Safe reinforcement learning by imagining the near future, 2021.

---

> > ### Comment · Reviewer_RuZi · 2023-08-14
> >
> > Thank you for the clarification and additional experiments. By viewing this, I have increased my score.

---

> > > ### Author Response · Authors · 2023-08-16
> > >
> > > Thank you very much for raising the score!

---

### Author Rebuttal · Authors · 2023-08-09

Thanks for the responses of all the reviewers! The experiment curve with variance is attached in the PDF. We only plot the first few
episodes to show the variance of each algorithm more clearly.

---

### Decision · Program_Chairs · 2023-09-21

**Decision:**

Accept (poster)

**Comment:**

The paper explores reinforcement learning with rigorous step-wise violation constraints, wherein a state-dependent, non-negative cost accumulates at each decision-making step. It establishes lower bounds for both regret and safety violations and presents a model-based algorithm that meets these lower bounds. An additional algorithm with a constant gap-dependent violation is also offered. The authors further examine the reward-free case, introducing an algorithm with a leading-term sample complexity that matches the state-of-the-art for cases without constraints. The paper also provides experiments to justify the theory.

**Reasons to Accept:**
The paper studies an important problem and the proposed stepwise violation constraint setting is novel. The theoretical results are reasonable and experiments are also adequate.  However, reviewers do find that the theoretical contributions are limited and can be improved to, e.g., zero constraint violation.